# Phospho-seq: integrated, multi-modal profiling of intracellular protein dynamics in single cells

John D. Blair[1,2], Austin Hartman[1,4], Fides Zenk[3,4], Philipp Wahle[3], Giovanna Brancati[3], Carol Dalgarno[1], Barbara Treutlein[3] & Rahul Satija[1,2] ✉

Cell signaling plays a critical role in neurodevelopment, regulating cellular behavior and fate. While multimodal single-cell sequencing technologies are rapidly advancing, scalable and flexible profiling of cell signaling states alongside other molecular modalities remains challenging. Here we present Phospho-seq, an integrated approach that aims to quantify cytoplasmic and nuclear proteins, including those with post-translational modifications, and to connect their activity with cis-regulatory elements and transcriptional targets. We utilize a simplified benchtop antibody conjugation method to create large custom neuro-focused antibody panels for simultaneous protein and scATAC-seq profiling on whole cells, alongside both experimental and computational strategies to incorporate transcriptomic measurements. We apply our work-flow to cell lines, induced pluripotent stem cells, and months-old retinal and brain organoids to demonstrate its broad applicability. We show that Phospho-seq can provide insights into cellular states and trajectories, shed light on gene regulatory relationships, and help explore the causes and effects of diverse cell signaling in neurodevelopment.

The ability to carefully regulate responses to external and internal signals is essential for proper cellular function. Maintaining precise control of signaling networks enables cells to achieve homeostasis in response to environmental changes and to progress through key developmental and functional transitions[1]. Signal transduction links the activation of signaling pathways to downstream changes in cellular chromatin[2], transcription[3], and translation[4], and is primarily regulated by changes in post-translational modifications[5]. For example, changes in phosphorylation can significantly alter receptor kinetics[6], enzymatic function[7], and transcription factor localization and activity[8] via conformational changes[9].

Cell signaling pathway activity is known to vary throughout neurodevelopment, and when disrupted, it can have significant effects on cell fate decisions. For instance, uncontrolled mTOR pathway activity caused by the loss of the upstream regulator TSC2 leads to a significant increase in astrogliosis, which is a hallmark of the neurodevelopmental disorder, tuberous sclerosis[10]. Additionally, changes in Wnt signaling have been associated with autism spectrum disorder[11]. To gain a deeper understanding of the causes and consequences of aberrant signaling, a crucial challenge is to connect heterogeneity in the activation of signaling pathways with broader changes in molecular state. Therefore, methods that can accurately measure and quantify phosphorylation states at single-cell resolution alongside additional molecular modalities offer substantial promise to improve our understanding of cellular activity and function.

Phospho-proteins are typically quantified in single cells using antibody-based methods including immunocytochemistry, flow cytometry[12], or CyTOF[13], which can be multiplexed to detect dozens of targets. Single-cell sequencing technologies offer an exciting opportunity to build upon these approaches, particularly with the recent introduction of multiomic technologies that enable the quantification of multiple modalities of information within the same cell[14–21]. For

[1]New York Genome Center, New York, NY, USA. [2]Center for Genomics and Systems Biology, New York University, New York, NY, USA. [3]ETH Zurich, Basel, Switzerland. [4]These authors contributed equally: Austin Hartman, Fides Zenk. ✉e-mail: rsatija@nygenome.org

example, CITE-seq[15], REAP-seq[19], DOGMA-seq[18], and TEA-seq[20] all stain cells with large panels of oligonucleotide-tagged antibodies against cell surface proteins in order to quantify cellular immunophenotypes alongside cellular transcriptomes, chromatin accessibility profiles, or both. While powerful, these technologies focus exclusively on the profiling of cell surface proteins. They are therefore widely applied to analyze hematopoietic samples, where well-characterized panels of cell surface proteins are associated with distinct cell states, but have limited utility in other contexts, including neurodevelopment.

Multiple pioneering approaches have built upon these methods, aiming to utilize single-cell multimodal technologies to profile intracellular and intranuclear proteins in diverse biological contexts. These include ASAP-seq[18], which introduces a set of fixation and permeabilization conditions that are compatible with chromatin accessibility profiling in whole cells. Additionally, inCITE-seq[21] and NEAT-seq[16] utilize specialized approaches for intranuclear protein profiling, which significantly reduce background signal originating from non-specific electrostatic interactions between oligonucleotide-conjugated antibodies and charged cellular components. Alternatively, one can fix, stain and sort cells into high and low populations of individual phospho-proteins, performing single-cell sequencing on the identified populations, as in INs-seq[22]. While each of these methods addresses key challenges, they perform profiling of small (3–7) intracellular panels due to a reliance on commercially conjugated antibodies. QuRIE-seq aims to profile larger panels, but requires custom instrumentation and is not compatible with primary cell samples[17]. Lastly, while NEAT-seq utilizes the 10x Multiome kit for trimodal nuclear profiling, no existing approaches can quantify intracellular proteins (including phosphorylation states), transcriptional output, and chromatin profiling in the same biological system.

Here we present Phospho-seq, a multi-modal single-cell workflow for quantifying cell signaling via phosphorylated cytoplasmic and nuclear proteins in conjunction with chromatin accessibility and gene expression levels (either directly measured experimentally, or integrated computationally[23]). By optimizing a broadly accessible antibody conjugation strategy[24], we designed custom panels to profile up to 64 intracellular proteins (including 20 phospho-states), and applied this workflow to both retinal organoids and brain organoids. We used the trimodal chromatin, RNA, and protein data to discover novel transcription factor-cis-regulatory element-gene associations including telencephalic and diencephalic lineage-specific relationships. We further used the phosphorylated protein data to identify lineage-specific patterns of Wnt and MAPK/ERK signaling, and to link these differences to upstream and downstream molecular networks. Overall, we demonstrate the utility of Phospho-seq for the high throughput discovery of interactions between cell signaling, gene regulation, and gene expression in neural tissue, opening the door for future discoveries in other cell types and tissues.

## Results

### Benchtop conjugation enables antibody panel customization in Phospho-seq

In developing Phospho-seq, we aimed to create a user-friendly, single-cell method to quantify proteins from the cell surface, cytoplasm, and nucleus alongside additional molecular modalities. To maximize utility, we aimed for the assay to 1) allow for maximum customizability by the user for which proteins to quantify, 2) rely only upon commercially available reagents and equipment, and 3) maintain the sensitivity and specificity of single-modality assays. Towards these goals, Phospho-seq combines aspects of previously established single-cell protein and chromatin accessibility quantification methods[16,18]. In brief, samples are dissociated into single-cells, fixed, permeabilized, hashed, stained for intracellular proteins with self-conjugated DNA-bound antibodies, and run through the 10X Genomics single-cell ATAC-seq protocol (Fig. 1A and Methods).

A major limitation of large-scale sequencing-based intracellular profiling is the lack of commercially available oligonucleotide-tagged antibodies. Premade antibody panels target immune cell surface antigens[25], while custom commercial antibody conjugation is often prohibitively expensive, especially for larger panels. To overcome this issue, we optimized a simple benchtop click-chemistry-based conjugation protocol[24] (Fig. 1B) to generate panels of uniquely-indexed oligonucleotide-conjugated antibodies. This approach is scalable, cost-effective (~$8/conjugation), and compatible with unconjugated commercial antibodies that are routinely used for immunofluorescence or flow cytometry, even those with carrier proteins (Fig. 1C, Fig. S1A, B). Phospho-seq is compatible with antibodies conjugated to different oligonucleotide sequences, including the widely used TotalSeq-A and TotalSeq-B sequences, through the use of a bridge oligonucleotide for capture on the gel beads[18].

During the optimization of our conjugation protocol, we found that the ratio of antibody to oligonucleotide and post-conjugation purification steps were crucial for minimizing nonspecific binding while maintaining a high recovery yield. Through titration experiments, we determined that adding 15 pmol of oligonucleotide per μg of antibody (equivalent to 2–4 copies of oligonucleotide per antibody molecule) was optimal (Fig. 1D, Fig. S1C). For post-conjugation purification, we found that two steps: an initial precipitation step using 40% ammonium sulfate[26], followed by 5–7 washes through a 50 kDa molecular weight cut-off (MWCO) filter, were necessary to reduce unbound oligonucleotide while preserving antibody yield (Fig. S1D).

To optimize and confirm that our Phospho-seq procedure could retain cell-to-cell differences in intracellular antibody levels, we utilized flow cytometry experiments on whole cells. We first confirmed that light fixation and gentle detergent-based permeabilization, as suggested by ASAP-seq[18], achieved the necessary balance between maintaining the structural integrity of the cell membrane while allowing the flow of unconjugated antibodies into the nucleus and cytoplasm (Methods, Fig. 1E). We also found that the addition of single-stranded DNA binding protein (SSB) to our antibody pool before staining, as pioneered for nuclear profiling with NEAT-seq[16], was essential to reduce background signal from oligonucleotide-conjugated antibodies. Combining these approaches, we utilized flow cytometry to successfully quantify clear SOX2 expression differences in heterogeneous mixtures of whole cells using an oligonucleotide-conjugated antibody (Fig. 1E and Fig S1E,F).

### Phospho-seq simultaneously quantifies phosphorylated, cytoplasmic, and nuclear proteins

To evaluate the full Phospho-seq workflow on whole cells, we first tested a small panel of both nuclear and cytoplasmic proteins on a heterogeneous cell line mixture. Our panel included antibodies against the transcription factors OCT4, SOX2, and GATA1, which are expected to be differentially expressed between K562 cells and induced pluripotent stem cells (iPSCs). We also aimed to quantify phosphorylated ribosomal protein S6 (pRPS6) expression, a readout of PI3K/AKT/mTOR pathway activation, along with antibodies quantifying total RPS6 levels. We exposed cells to either epidermal growth factor for 1 h (EGF, activating) or PX-866 for 4 h (inhibiting) to modulate pathway activation, and utilized cell surface hashing antibodies[27] to perform multiplexed profiling of cells in resting, activated, or inhibited conditions.

We found that Phospho-seq was able to quantify nuclear, cytoplasmic (including phosphorylated), and cell surface proteins alongside chromatin accessibility. Dimensional reduction of scATAC-seq profiles successfully discriminated the two cell lines and was concordant with the cell hashing-based demultiplexing. (Fig. 1F and S1G, H). Antibody-derived tags (ADTs) for the defining canonical transcription factors OCT4, SOX2 (iPSCs) and GATA1 (K562) showed differences between the two cell types (OCT4 Staining Index (SI)I:1.22; $p < 0.001$, SOX2 SI:0.56; $p < 0.001$, GATA1

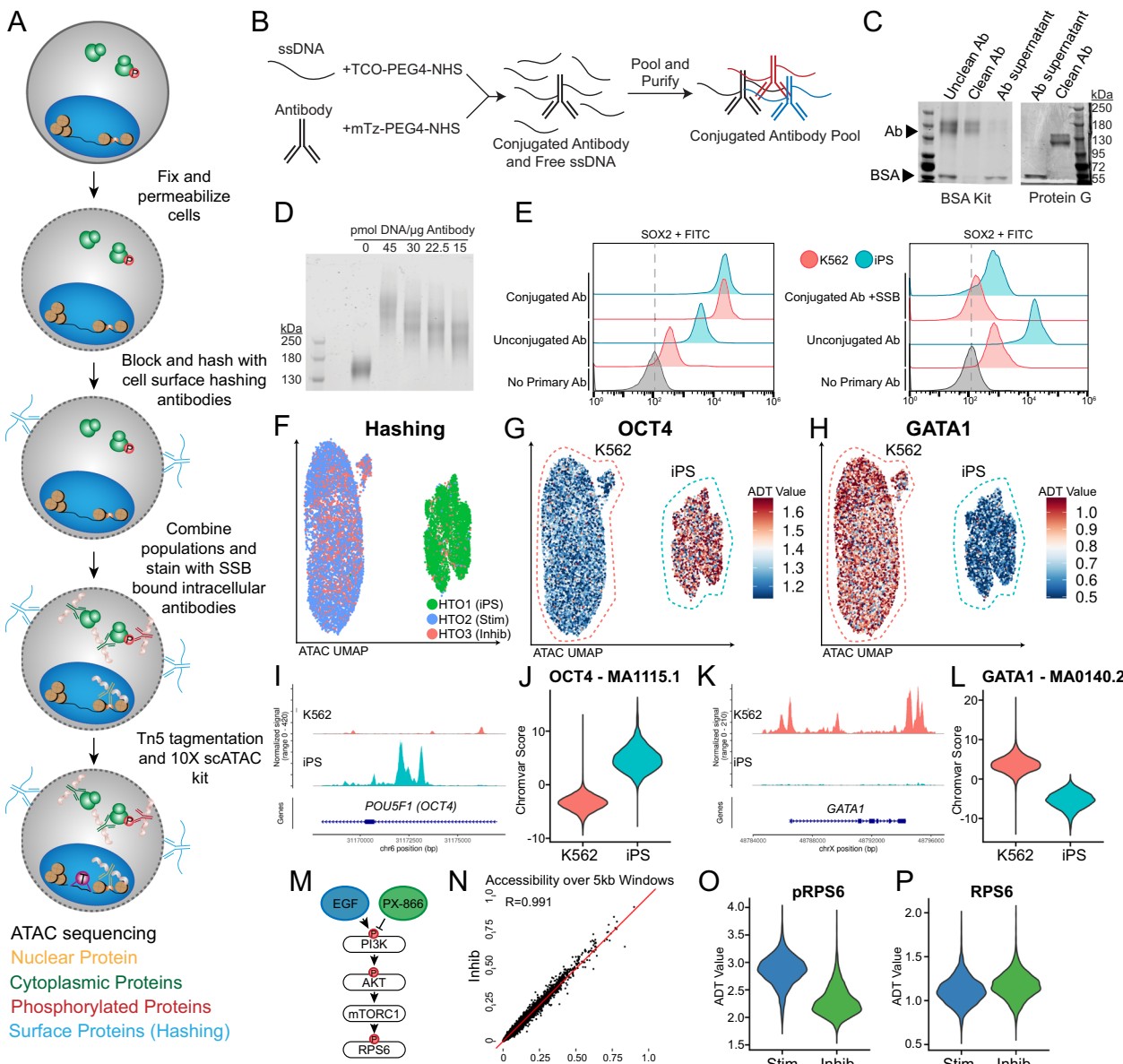

**Fig. 1 | Phospho-seq experimental workflow and pilot experiment. A** Schematic of Phospho-seq workflow. **B** Schematic of antibody conjugation procedure. **C** Protein gel results of two antibody (Ab) purification methods using the Abcam BSA Removal Kit (left panel) and Promega Magne Protein G beads (right panel). This experiment was performed once. **D** Protein gel of mTz-PEG4-NHS labeled antibodies incubated with different quantities of TCO-PEG4-NHS labeled ssDNA tags. This experiment was performed once. **E** Flow cytometry plots of K562 and iPS cells stained with unconjugated and conjugated SOX2 antibodies (left panel) and cells stained with unconjugated and conjugated + single-stranded DNA binding protein SOX2 antibodies (right panel) with unstained controls. **F** UMAP representation from scATAC-Seq of K562 and iPS cells colored by demultiplexed HTOs assigned to each cell. **G** UMAP representation of K562 and iPS cells colored by normalized ADT values for OCT4. **H** UMAP representation of K562 and iPS cells colored by normalized ADT values for GATA1. **I** Coverage plot of chromatin accessibility of K562 and iPS cells at the POU5F1 (OCT4) genomic locus. **J** Violin plot of chromVAR scores for the OCT4 TF binding motif (MA1115.1) in K562 and iPS cells. **K** Coverage plot of chromatin accessibility of K562 and iPS cells at the GATA1 genomic locus. **L** Violin plot of chromVAR scores for the GATA1 TF binding motif (MA0140.2) in K562 and iPS cells. **M** Schematic of PI3K/AKT/mTORC1 pathway activation and repression paradigm used in this experiment. **N** Scatter plot of pseudobulked chromatin accessibility data in 5 kb windows across the genome comparing inhibited K562 cells with stimulated K562 cells. Red line indicates perfect correlation between the two conditions. **O** Violin plot of normalized pRPS6 values in stimulated (Stim) and inhibited (Inhib) K562 cells. **P** Violin plot of normalized RPS6 values in stimulated (Stim) and inhibited (Inhib) K562 cells.

SI:1.29; $p < 0.001$) (Fig. 1G, H and Fig S1I). These profiles were also concordant with the chromatin accessibility landscape at each protein's genomic locus (Fig. 1I, K and S1J) and genome-wide transcription factor motif activity estimates as quantified by chromVAR[28] (Fig. 1J, L and S1K).

Moreover, while changes in PI3K/AKT/mTOR pathway (Fig. 1M) activity did not drive genome-wide changes in chromatin accessibility (Fig. 1N), we observed a clear increase in pRPS6 levels within stimulated cells compared to inhibited cells (Fig. 1O) independent of total

RPS6 levels (Fig. 1P). This reflects a biological context where phosphorylation measurements are highly informative for distinguishing cellular states even when genome-wide modalities cannot. We also observed higher phosphorylation of the nuclear transcription factor STAT3 in iPSCs compared to K562 cells, and found that only pSTAT3 levels (as opposed to total protein levels) correlated with STAT3 transcription factor activities (Fig S1L). We conclude that we can therefore measure phosphorylation states for both cytoplasmic and

nuclear activities, and that phosphorylated protein levels are more reflective of cellular state and transcription factor activities compared to total protein levels.

We further observed that Phospho-seq was capable of quantifying more subtle differences, even within the same cell type. Epigenetic differences between iPS donor cell lines are frequently observed and often lead to biased differentiation tied to cell signaling differences[29]. In our dataset, we observe the segregation of the three iPSC donors used in this experiment based on chromatin accessibility (Fig S1M) as well as an enrichment of pRPS6 signal in one donor in particular (Fig. S1N). These types of observations may be highly informative when assessing the differentiation capacity of different iPSC lines.

While the previous analyses qualitatively demonstrate the multimodal capabilities of Phospho-seq, we next aimed to quantitatively assess the protein measurements from Phospho-seq, and to benchmark them against alternative technologies. We leveraged quantitative metrics—originally developed for flow cytometry—that are widely used to perform relative comparisons of protocols and experimental conditions - in particular, the staining index (SI) leverages both the median and variance of protein levels between two populations[30]. We performed two quantitative benchmarking experiments that leveraged this metric.

First, we performed both flow cytometry and Phospho-seq on two cell lines, K562 and HEK. We used antibodies against a panel of four cytoplasmic proteins (ELA2, VIM, APPL1, and IFITM1), a surface protein (CD55), and a nuclear protein (GATA1) that we expected to show abundance differences between the two cell lines (with the exception of ELA2, a negative control). Both technologies exhibited upregulation of GATA1, CD55, and IFITM1 in K562 cells, and VIM and APPL1 in HEK cells (Fig. S2A,B). While Phospho-seq exhibited a higher SI on average (mean SI = 1.26 vs. 0.74. for flow cytometry), we did not identify a statistically significant difference between the two ($p = 0.375$; Fig. S2C), suggesting that our Phospho-seq data is comparable to gold-standard approaches. These findings are consistent with previous work that suggests that the dynamic range of sequencing-based protein quantification matches and may in fact exceed traditional colorimetric cytometry, as the sequencing-based measurements are digital, contain unique molecular identifiers, and may be less affected by saturation of the detector[15]. Moreover, we quantified each protein in serial independent stains for flow cytometry (removing any possibility of spectral overlap), while the inherent multiplexing of Phospho-seq enabled all proteins to be quantified in a single experiment.

Second, we quantitatively benchmarked Phospho-seq with ASAP-seq[18], the only other technology that is described as being capable of simultaneous intracellular protein and chromatin profiling at single-cell resolution. Here, we took a complementary approach where we obtained an estimate of 'ground truth' protein levels from flow cytometry, ran the same sample through the sequencing-based assay, and compared the two results. To accomplish this, we stained cells with a mixture of pRPS6 antibodies, enabling them to be first profiled by FACS, and subsequently by sequencing. Using chemical stimulation (Methods) we generated a heterogeneous population of K562 cells, divided this into four gated populations based on pRPS6 measurements from flow cytometry, and subsequently profiled each gate with Phospho-seq, ASAP-seq and flow cytometry (Fig. S2D–J). Comparing the mean expression of pRPS6 between gated populations in Phospho-seq, we observed quantitative agreement ($R^2 = 0.90$), highlighting the consistency of Phospho-seq with ground truth measurements (Fig. S2L). With ASAP-seq, we observed a weaker correlation ($R^2 = 0.12$; Fig. S2K). We conclude that Phospho-seq exhibits best-in-class performance for technologies measuring protein and chromatin at single-cell resolution, and that the dynamic range of the assay matches flow cytometry-based methods that lack Phospho-seq's multiplexing and multimodal capabilities.

## Intracellular protein staining highlights cell type differences in human retinal organoids

To explore the applicability of Phospho-seq to neural tissue, we utilized human iPSC-derived retinal organoid models (Fig. 2A), which have been shown to recapitulate retinal development, producing a heterogeneous cell population including photoreceptors, glia, and epithelium[31]. We conjugated a custom panel of 46 antibodies, the majority previously used in an iterative indirect immunofluorescence imaging (4i) study[32], consisting of mature retinal cell type markers as well as developmentally relevant transcription factors and one phospho-signaling antibody, pRPS6 (Supplementary Data 1). Running Phospho-seq with this panel on 34-week-old retinal organoids while filtering for doublets with cell hashing[27] produced a dataset with 8136 cells.

Utilizing dimensionality reduction and unsupervised clustering, we identified 16 cell clusters (Fig. S3A). While we could assign broad cell type labels using gene activity scores calculated from chromatin accessibility within gene bodies and promoters[33] (Fig. 2B and S3B), more fine-grained annotation is challenging for chromatin accessibility measurements in the absence of transcriptomic data. We found that ADTs were heterogeneously detected across differently assigned cell types, identifying markers of progenitors and glia (SOX9, VIM), photoreceptors (RCVRN, DNM1), Rods (RHO), and retinal pigment epithelium and glia (RLBP1) (Fig. 2C, D and Fig. S3C, D). Elevated protein marker expression detection was further supported by enriched chromatin accessibility in the same cell types for both intranuclear and intracellular proteins (Fig. 2C, D, and Fig. S3C). Correlating individual TF ADT values with accessibility of all possible binding motifs, as determined by chromVAR[28], we found that for 6 out of the 10 TFs profiled, the correct binding motif was in the 80th rank-correlation percentile or above (Fig. S3E, F). For these instances where ADT capture and canonical motif accessibility are highly rank-correlated for a TF, this is evidence of direct, functional regulation of chromatin accessibility by the target protein.

Focusing on phosphorylation, we observed a large, previously unreported difference in pRPS6 signal between rods and cones (Fig. 2E), which we validated through immunofluorescent imaging (Fig. 2F, G and Fig. S3G). Photoreceptors are among the most metabolically demanding cell types in the body, requiring high levels of mTOR signaling, which in turn phosphorylates RPS6, to properly function[34]. Among photoreceptors, cones require more energy than rods due to their more consistent depolarized state[35] and the relative levels of pRPS6 we observe between these two cell types reflect this. This observation demonstrates the utility of Phospho-seq to discover signaling differences between cell types.

## Extending Phospho-seq to incorporate transcriptomic measurements

Our Phospho-seq protocol utilizes the 10x scATAC-seq kit to generate scalable and high-quality chromatin accessibility profiles but lacks transcriptomic measurements, which are highly valuable for fine-grained cell annotation and gene regulatory network reconstruction. We therefore considered both experimental and computational strategies to incorporate transcriptomic profiles into the Phospho-seq assay.

To experimentally obtain transcriptome information in a Phospho-seq experiment, we employed the 10X scATAC + RNA multiome kit, in place of the scATAC kit, and applied the Phospho-seq-multi workflow to retinal organoid samples (Methods) (Fig. 3A). We successfully generated libraries for all modalities and recovered 1474 cells from an input of 5000 with a median of 866 RNA UMIs, 8326 ATAC fragments and 39 ADT UMIs per cell (Supplementary Data 5). Dimensionality reduction and clustering Phospho-seq-multi data using weighted-nearest neighbors[25] revealed 7 clusters, which we assigned identities based on their RNA expression (Fig. 3B and Fig. S4A). We verified the expected accessibility and expression patterns of both cell-

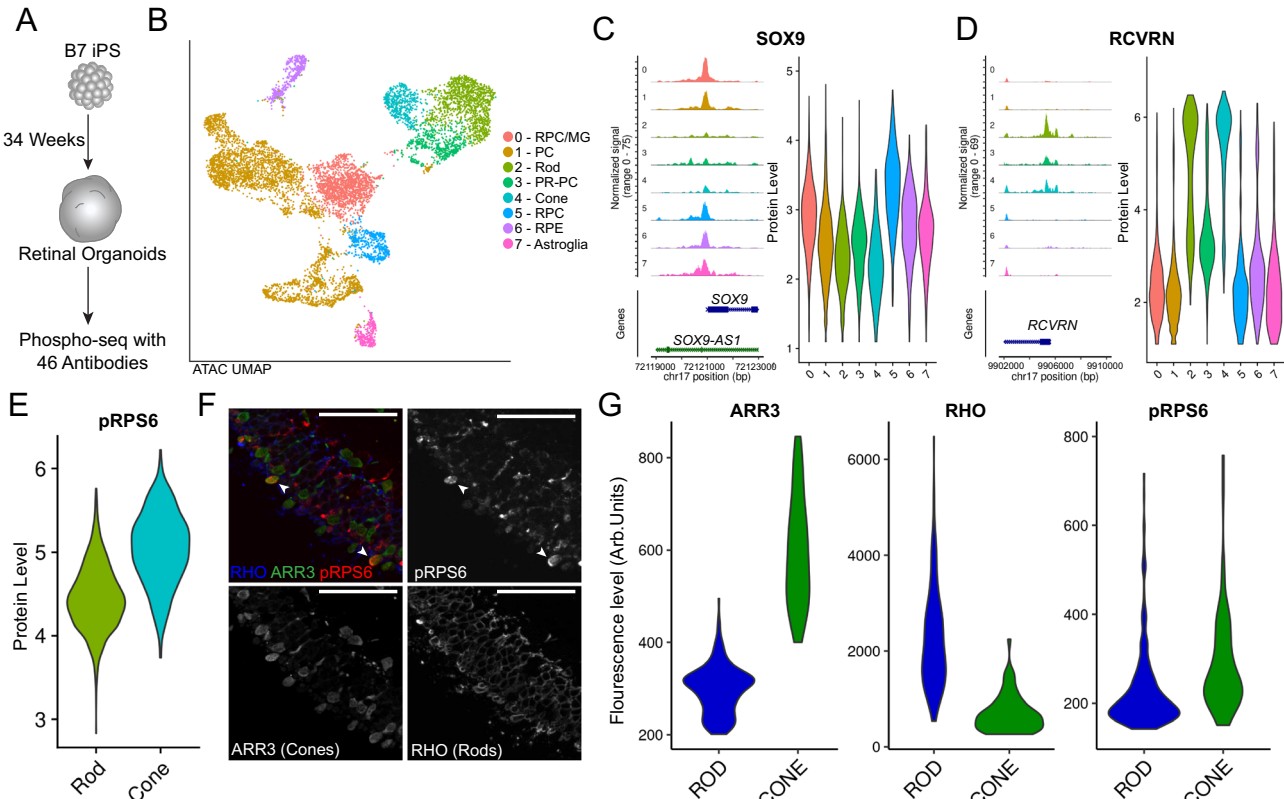

**Fig. 2 | Phospho-seq on retinal organoids. A** Schematic of retinal organoid differentiation. **B** UMAP representation of cells and cell type assignments based on the ATAC-seq modality in Phospho-seq. RPC retinal precursor cell, MG müller glia, PC precursor cell, PR-PC photoreceptor precursor, RPE retinal pigment epithelium. **C** Coverage and violin plots of the gene promoter and protein level respectively of nuclear protein SOX9. Color and order are the same as in (**B**). **D** Coverage and violin plots of the gene promoter and protein level respectively of cytoplasmic protein RCVRN. Color and order are the same as in (**B**). **E** Violin Plot of pRPS6 ADT quantification in Rods and Cones. **F** Confocal images of 33-week-old retinal organoids harvested and immunostained with antibodies against the cone marker ARR3 (green), Rod marker RHO (blue), and pRPS6 (red) with examples of high-pRPS6 cones indicated by arrowheads. Scale bars are 50 μm. This experiment was performed once. **G** Quantification of ARR3, RHO, and pRPS6 in 240 Rods and 64 Cones from (**F**).

type specific enhancers as well as protein markers from the ADT profiles, including canonical ADTs like RCVRN (photoreceptors), VIM (Precursors) and RLBP1 (RPE and glia) (Fig. S4B), demonstrating the capability of Phospho-seq to simultaneously pair intracellular protein profiles alongside both transcriptomic and chromatin accessibility measurements.

While this dataset was informative and enabled cell type identification from transcriptomic profiles, the simultaneous acquisition of multiple modalities came at a substantial cost in data quality and throughput when compared to either our existing Phospho-seq data, or scRNA-seq on retinal organoids[32]. Specifically, we observed a 70% reduction in molecular sensitivity for scATAC-seq profiles and a 91% reduction for gene expression profiles (Fig. S4C, D). We further observed a significant reduction in the staining index for both intracellular and intranuclear ADT levels, with a range of 45–70% reduction between three pairs of cell types (Fig. 3C and Fig. S4E and Supplementary Data 2). Lower ADT sensitivities may be mitigated through the use of direct capture (i.e. TotalSeq-A) ADT oligos, as in NEAT-seq[16] rather than indirect capture oligos (TotalSeq-B) due to the differing chemistry between the scATAC kit and the multiome kit. Regardless, these data are consistent with previous studies[20] that exhibit lower permodality molecular sensitivity when using the 10x multiome kit, and demonstrate the inherent challenges for multimodal technologies to generate high-quality data from each measurement type.

We, therefore, explored an alternative approach aiming to integrate Phospho-seq and scRNA-seq reference datasets using our recently introduced 'bridge integration' procedure (Fig. 3D)[23]. We have previously shown that this workflow can successfully integrate distinct

modalities collected in different experiments by leveraging a separately obtained multiomic dataset as a 'bridge', even if the bridge dataset has reduced technical quality. The bridge integration procedure can successfully integrate data for both discrete cell types as well as continuous developmental trajectories, but requires that the multiomic dataset is biologically representative (i.e. inclusive of all cell types and states) of the single-modality datasets. In our study, the previously published bridge and reference datasets were generated from the same set of samples[32] (Fig. S4F, G), thereby satisfying this assumption and demonstrating that integrating Phospho-seq datasets does not require the generation of additional multiomic data if such data already exists.

Applying this workflow, we used bridge integration to annotate each Phospho-seq profile based on published labels (Fig. 3E). We found that these transferred annotations were consistent with those originally derived from gene activity scores, but increased the interpretability of annotation. For example, Phospho-seq cells that were annotated as Immature PRs fell into four categories after bridge integration: BCs, AC/HC, Early PRs, and non-retinal cells (Fig. 3F). After we performed bridge integration, we retrospectively identified differences in the chromatin accessibility profiles across high-resolution subsets that supported our transferred annotations. For instance, there were previously unobserved accessibility differences at canonical markers between progenitors and differentiated cells for Bipolar and Amacrine/Horizontal Cells (Fig. S4H). Moreover, we obtained high prediction scores (70% >0.50) for all cell types with the exception of cells that were not sufficiently represented in the bridge data, like Photoreceptor Precursors (PR-PC) or non-retinal cells (Fig. S4G, I).

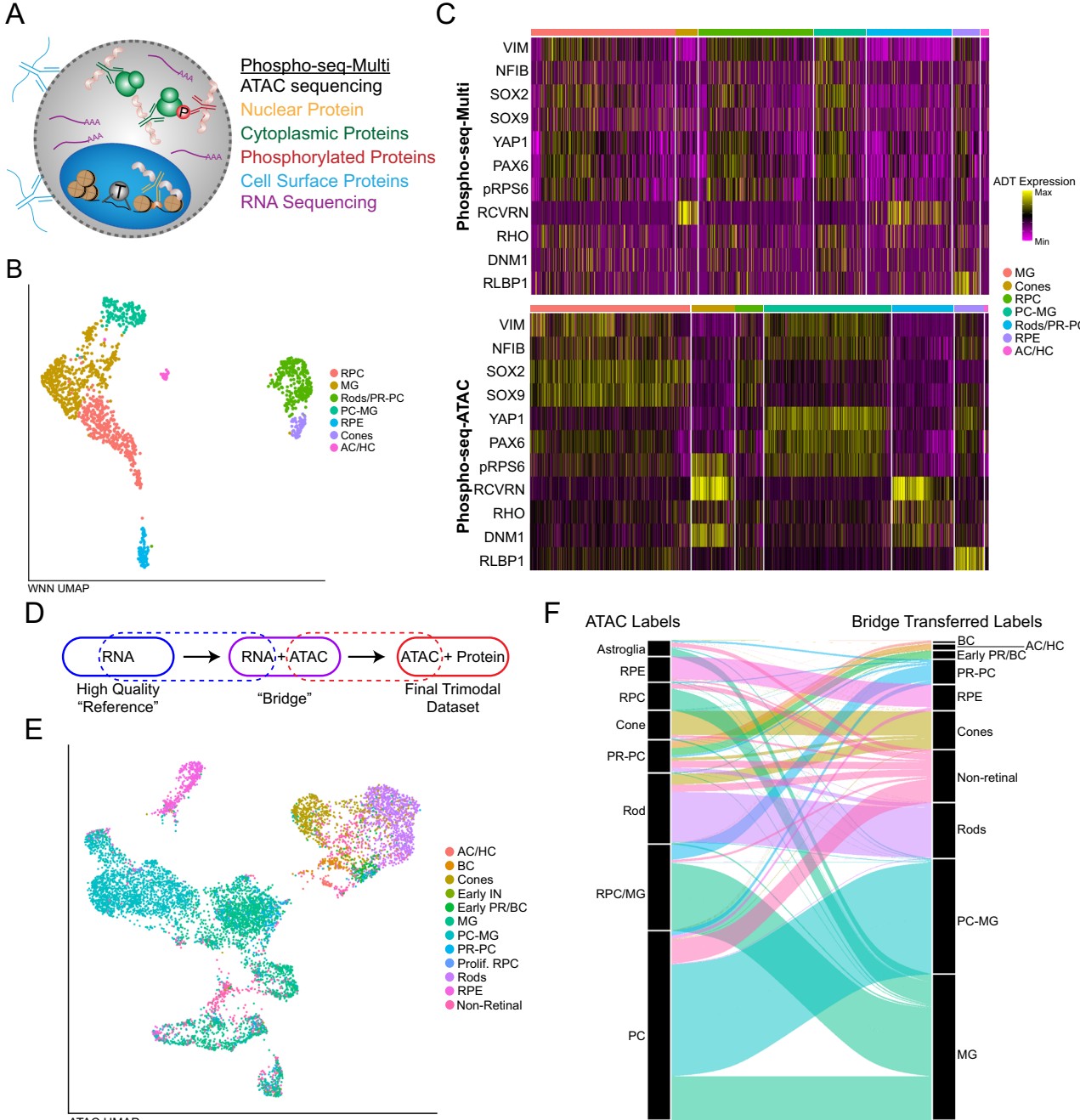

**Fig. 3 | Extending Phospho-seq with transcriptomic measurements.**
**A** Schematic of Phospho-seq using the 10x multiome kit. **B** Weight-nearest neighbors UMAP of Phospho-seq-Multi experiment labeled by cell type. AC/HC amacrine cells/horizontal cells. **C** Heatmap comparison of 11 ADTs between Phospho-seq experiments using either the Multiome kit or the ATAC-seq kit. **D** Schematic of bridge integration procedure. **E** UMAP representation of retinal organoid cells based on ATAC-seq modality with cell type assignments from bridge integration. BC bipolar cells. IN intermediate. **F** Alluvial plot demonstrating cell label transfer when using bridge integration.

We therefore utilized our integrated dataset to impute transcriptome-wide expression profiles for each Phospho-seq cell. We confirmed that these imputed profiles maintained high rank-correlation for TF motif accessibility, as well as measured ADT levels for the same gene (Fig. S4J–L). These results confirm the applicability of bridge integration to obtain a high-quality RNA modality with Phospho-seq. We conclude that, while it is possible to use the Phospho-seq workflow to perform experimental trimodal profiling, this comes with a cost in data quality and thus integrated analysis may result in improved data quality for multiple modalities.

## Intracellular protein staining highlights cell type differences in human brain organoids

Finally, we chose to apply Phospho-seq to uncover signaling patterns in early neurodevelopment. For this, we used human iPSC-derived brain organoid models, which have provided valuable insights into neurodevelopmental processes that are otherwise difficult to study in humans[36,37]. We first designed and conjugated a custom panel of 64 antibodies comprising both neurodevelopmentally relevant transcription factors, as well as an expanded set of 40 paired cell-signaling antibodies (Fig. 4A and Supplementary Data 1). While previous studies

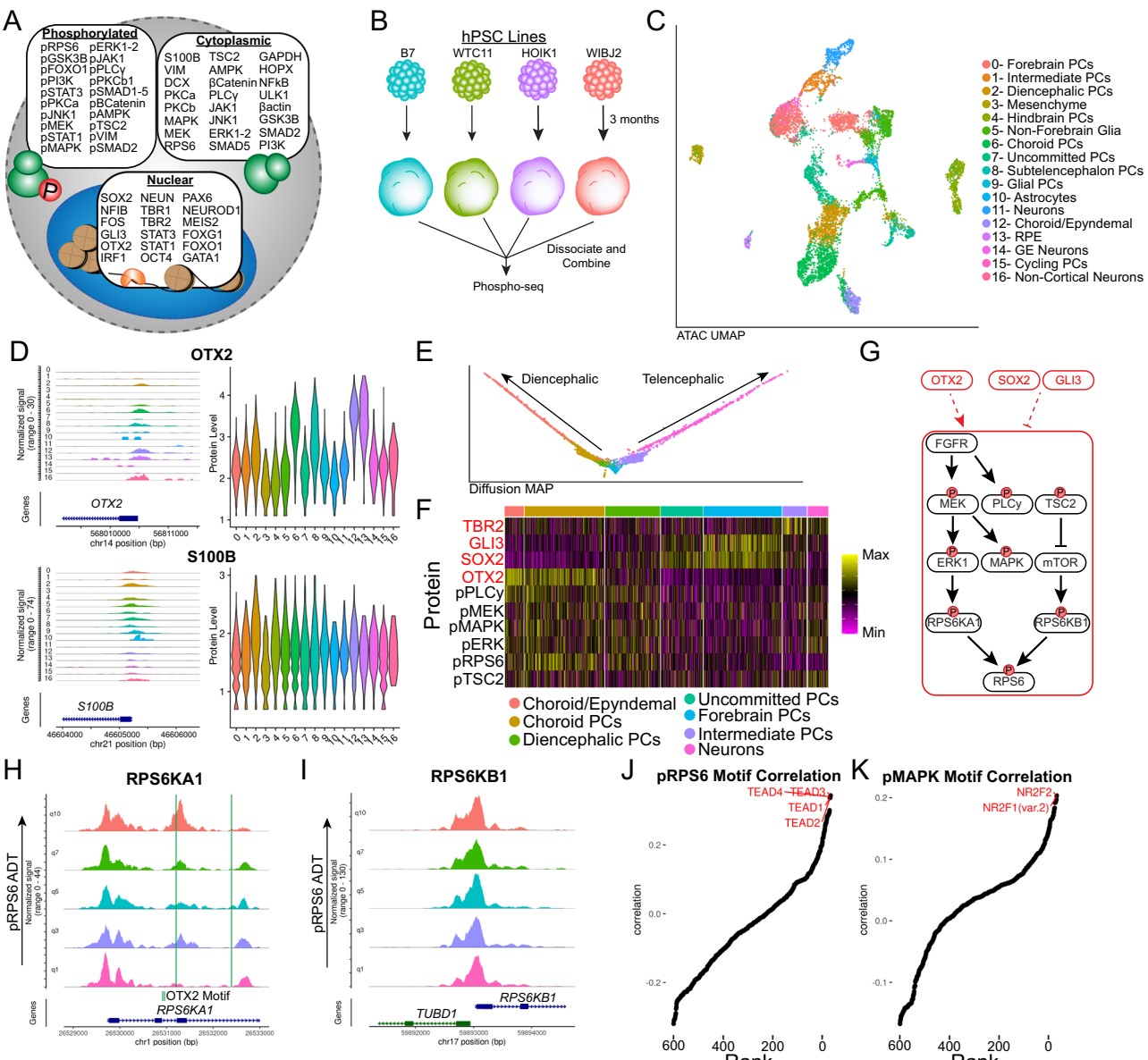

**Fig. 4 | Phospho-seq on brain organoids. A** Schematic of antibody panel used in brain organoid Phospho-seq experiment and the cellular compartment of the target protein. **B** Schematic of brain organoid differentiation. **C** UMAP representation of cells and cell type assignments based on the ATAC-seq modality in brain organoid Phospho-seq. GE ganglionic eminence. **D** Coverage and violin plots of the gene promoters and protein levels respectively of nuclear protein OTX2 and cytoplasmic protein S100B, color and order are same as in (**C**). **E** Diffusion MAP plot of diencephalic and telencephalic differentiation trajectories. Cell-type colors correspond to legend in (**C**). **F** Heatmap of ADT expression of proteins and phospho-proteins associated with MAPK/ERK and mTOR signaling. **G** Schematic of MAPK/ERK signaling. **H** Coverage plot of RPS6KA1 split into quantiles of pRPS6 levels across telencephalic and diencephalic differentiation trajectories with OTX2 binding motifs indicated. **I** Coverage plot of RPS6KB1 split into quantiles of pRPS6 levels across telencephalic and diencephalic differentiation trajectories. **J** Rank-correlation plot showing correlation between pRPS6 levels and motif accessibility across the whole Phospho-seq dataset. The top hits are indicated in red. **K** Rank-correlation plot showing correlation between pMAPK levels and motif accessibility across the whole Phospho-seq dataset.

have profiled chromatin and transcriptomic modalities from these models[37,38], we reasoned that our Phospho-seq panel could illuminate relationships between cell signaling pathways, transcription factors, and gene regulatory elements. We separately generated 3-month-old organoids from four iPS donors, using a well-established protocol for unguided brain organoid differentiation (Fig. 4B)[37,39]. We performed cell hashing for doublet detection[27] and used genetic profiling[40] for donor demultiplexing, subsequently profiling 9,028 cells using Phospho-seq (Fig. S5A).

Utilizing dimensionality reduction and unsupervised clustering, we identified 10 clusters representing heterogeneity both in neurodevelopmental lineage and donor identity (Fig. S5B) assigning broad

cell type labels using gene activity scores[33] (Fig. S5C). To perform bridge integration for this dataset, we generated a new scRNA-seq reference dataset with 19,280 cells and a multiomic bridge dataset with 4958 cells, all from the same samples (Fig. S5D, E). Using canonical gene expression markers, we manually annotated the reference dataset (Fig. S5F) and transferred these labels onto the Phospho-seq dataset (Fig. S5G). We obtained high prediction scores (95% >0.5) for all cell types with the exception of astrocytes, which are one of the most rare cell populations in this system and were not sufficiently represented in the bridge data (Fig. S5H).

We found ADTs that were heterogeneously detected across differently assigned cell types, identifying markers of forebrain

progenitors (SOX2, GLI3), diencephalic cells (OTX2), progenitors and glia (VIM, S100B), and intermediate progenitors (TBR2), again finding that elevated protein marker expression was further supported by enriched chromatin accessibility patterns in the same cell types for both intranuclear and intracellular proteins (Fig. 4D and S6A). We observed that 45 out of 64 proteins exhibited differential expression between at least one pair of clusters (Fig. S6B) and that most ADTs had high rank-correlation with their corresponding imputed RNA values compared to all other imputed RNAs (50% >80th percentile rank-correlation) (Fig. S6C).

We leveraged our integrated dataset to explore the relationship between chromatin accessibility, gene transcription, and protein levels across forebrain development. To do this, we first constructed a developmental trajectory across 1,578 cells spanning from progenitors to neurons (Fig. S7A)[41,42]. While *SOX2* gene expression decreased at the initial stages of differentiation, we observed a developmental lag in the decrease of downstream modalities including SOX2 ADT levels and SOX2 transcription factor activity as estimated by chromVAR (Fig. S7B). We identified a specific stage of the developmental trajectory where protein and RNA levels were discordant and confirmed that in these cells, only SOX2 protein levels (and not RNA levels) were correlated with chromVAR scores (Fig. S7C). SOX2 is a well-established negative regulator of neuronal development[43], and this analysis enabled us to identify genes whose downregulation preceded SOX2 protein downregulation, and are therefore unlikely to be direct or downstream targets of SOX2 itself, and vice-versa (Fig. S7D–F). We conclude that integrated Phospho-seq data can perform multimodal characterization of the distinct temporal patterns and gene regulatory relationships that drive cellular dynamics.

## Observing signaling network activity with Phospho-seq

One goal in developing Phospho-seq was to better characterize differences in signaling pathway activation across cell states and to connect these changes to gene regulatory networks. In this dataset, we observed an increase in phosphorylation of members of the MAPK/ERK pathway in the diencephalic lineage compared to the telencephalic lineage (Fig. 4E, F, Fig. S8A), which we validated through immunocytochemistry in organoids of the same donor and age (Fig. S8B–E). Of particular interest, we found that pRPS6 was heterogeneously expressed across cell types. While pRPS6 is often utilized as an indicator for mTOR pathway activity, RPS6 can actually be phosphorylated by multiple kinases from different pathways—including the ERK pathway via RPS6KA1 or the mTOR pathway via RPS6KB1[44] (Fig. 4G). We found that our Phospho-seq data could help us better understand determinants of RPS6 phosphorylation. We observed that chromatin accessibility at the *RPS6KA1* gene correlated with pRPS6 levels while accessibility at *RPS6KB1* was not correlated with pRPS6 levels (Fig. 4H, I). While the kinase activity of these proteins is determined by upstream phosphorylation, the variability of pRPS6 levels appears to be in part driven by variation at the *RPS6KA1* locus, indicating a more prominent role for ERK signaling over mTOR signaling in driving variable phosphorylation of pRPS6 in the diencephalic lineage.

As pRPS6 does not directly bind DNA, instead working to regulate gene expression through translation[45], we aimed to determine which TFs worked in concert with pRPS6 to achieve this regulation. When applying our ADT/motif rank correlation analysis, we identified a clear association between pRPS6 levels with multiple TEA/ATTS domain (TEAD) motifs in both brain organoids (Fig. 4J) and retina (Fig. S8F). TEAD transcription factors are the effectors of YAP-TAZ/Hippo signaling pathway, which is known to work together with MAPK/ERK signaling to regulate cell size[46] in cancer cells. Similarly, we found that pERK and pMAPK ADT levels were associated with the accessibility of NR2F1 and NR2F2 motifs (Fig. 4K), representing an important neurodevelopmental regulator whose RNA expression was also elevated in cells with higher activation of MAPK/ERK signaling[47,48] (Fig. S8G).

We further applied Phospho-seq to discover underlying signaling differences between related but distinct cell types. Heterogeneity between glial cells during development is indicative of future identity and function[49]. In this dataset we observed multiple glial subtypes (Fig. S9A) expressing canonical glial marker genes including Oligo-dendrocyte PCs (*OLIG1*+), Glial PCs (*SLC1A3*+), and two subtypes of *AQP4*+/*GFAP*+ astrocytes defined by *S100B*+ or *GLI3*+(Fig. S9B). While some signaling markers (pRPS6, pSTAT3) were unchanged across populations, pPLCγ ($p < 0.01$) and pAMPKa ($p < 0.05$) were elevated in S100B+ glia and depleted in SLC1A3+ glia (compared to S100B+ glia) respectively (Fig. S9C). These signaling pathways have been previously shown to reflect functional characteristics of cells, including energy usage (pAMPKa)[50] and calcium signaling (pPLCγ)[51]. As these functional characteristics are known to vary across both cell types and brain regions, we reference-mapped these cells with an scRNA-seq atlas of first-trimester developing human brains[52], finding four groups of glial cells with high mapping scores that expressed similar markers (Fig. S9D). Interestingly, we found that subcategories of glial cells we identified matched to different regions in the developing brain: *GLI3*+ glia and glial PCs were derived from the forebrain and telencephalon, and that *S100B*+ glia were from an equal distribution of non-telencephalic origin (Fig. S9E). Future in-vivo experiments will reveal whether these signaling differences replicate in vivo, and how they influence cellular decision-making and fate.

## Identification of transcriptional co-factors through ADT-motif correlation

We observed a rank-correlation of the 80th percentile or higher in 7 out of 16 TFs profiled (Fig. S10A). Looking closer, the OTX2 motif itself was the second-highest hit for OTX2 (Fig. S10B), with the other top hits having the same core motif. We observed similar results for SOX2 (Fig. S10C) and concluded that Phospho-seq data can help to identify and discover bona fide links between cellular protein levels and DNA sequences that influence gene expression. SOX2 and OTX2 are known to exhibit pioneer factor activity, where their binding alone can open heterochromatin[53,54].

As not all TF have the ability to act as pioneers and independently regulate chromatin state, we were also interested in further exploring cases where we did not observe strong correlations between intracellular protein levels and canonical motif accessibility. In these cases, we expect that TFs may work alongside additional co-factors, which we could learn from Phospho-seq data. For example, we observe that GLI3 protein levels showed only a weak correlation with the accessibility of the canonical GLI3 binding motif, but instead were associated with motifs from the homeodomain-2 (HD/2) family[55], which are known to affect neurodevelopment[56] (Fig. 5A). While this could in principle reflect non-specific antibody binding, we found that GLI3 protein exhibited high correlation with *GLI3* RNA levels (Fig. 5B; $R^2 = 0.86$), demonstrating that our GLI3 protein measurements are associated with the correct target.

We therefore hypothesized that other co-factors, such as those in the HD/2 family, may partner with GLI3 to regulate chromatin accessibility. Under this model, we propose that GLI3 does not individually act as a pioneer factor, but instead jointly regulates a subset of ATAC-seq peaks enriched for HD/2 family motifs. To test this model, we considered two additional data sources from the same neuronal organoid model as our Phospho-seq dataset[37]: a bulk GLI3 CUT&Tag dataset measuring genome-wide binding profiles, and a paired scRNA-seq + scATAC-seq dataset (10x multiome) that included both WT and *GLI3* KO cells. While CUT&Tag data revealed 331,131 GLI3 binding sites across the genome, only a small percentage (5108, 1.5%) of these were associated with differential accessibility (DA) upon *GLI3* perturbation. These DA regions were clearly enriched (fold-enrichment: 2.11) for GLI3 motifs over the background but even more strongly for HD/2 motifs (median fold-enrichment across HD/2 motifs: 3.02)(Fig. 5C), consistent

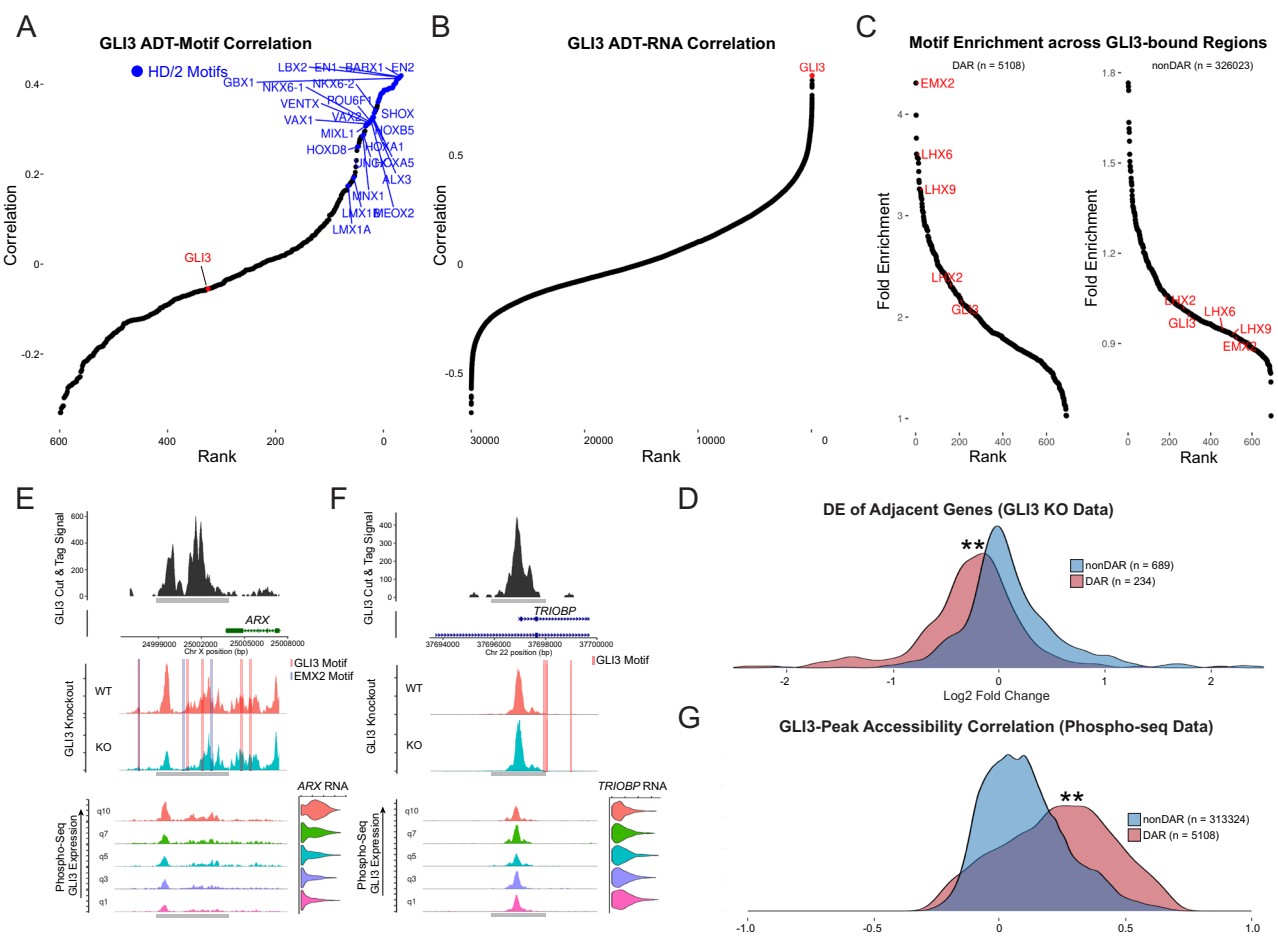

**Fig. 5 | Identification of GLI3 transcriptional co-factors. A** GLI3 protein levels correlate with the accessibility of HD/2 motifs (blue), but not the canonical GLI3 motif (red). Shown is a rank-correlation plot of all ADT/motif correlations. **B** GLI3 protein levels correlate with GLI3 RNA levels. Rank-correlation plot shows GLI3 ADT correlation with all genes. **C** Upon GLI3 perturbation, differentially accessible regions (DAR) are enriched for HD/2 motifs. Shown is the Rank-Fold Enrichment plot of motifs enriched in differentially accessible GLI3-bound regions in GLI3 knockout vs WT (left) and motifs enriched in non-differentially accessible regions (right). **D** Upon GLI3 perturbation, only genes linked to DAR are enriched for expression changes, which are predominantly negative as expected. Shown is a density plot of the fold change of genes linked to DAR and non-DAR. ** =

$p = 3.89 \times 10^{-23}$ using a two-sided t-test. Phospho-seq data in WT cells predicts which regions will respond to functional perturbation. **E**, **F** show two exemplary loci, both bound by GLI3 in CUT&Tag data. The region in (**E**) contains HD/2 and GLI3 motifs. In the GLI3 KO data, the region is differentially accessible (middle), and in the Phospho-seq data, the accessibility is correlated with GLI3 protein expression. The region in (**F**) is also bound by GLI3 but contains no HD/2 motifs, is not associated with GLI3 ADT levels in Phospho-seq, and is not DAR upon GLI3 perturbation. Density plot in (**G**) shows that this pattern holds genome-wide: the correlation between peak accessibility and GLI3 ADT levels in Phospho-seq data discriminates responsive (DAR) and non-responsive (nonDAR) peaks in GLI3 KO cells ** = $p = 2.2 \times 10^{-16}$ using a two-sided t-test.

with our Phospho-seq data. In addition, in the absence of chromatin changes, genes located adjacent to GLI3 binding sites showed no enrichment for differential expression after GLI3 perturbation, in contrast to the expression changes observed in DAR-adjacent genes (median average log2 fold-change: 0.04 vs −0.22; $p < 0.001$) (Fig. 5D−F; Supplementary Data 3). These findings suggest that GLI3-binding alone is insufficient to regulate chromatin accessibility and gene expression, and therefore, that CUT&Tag data is a poor predictor of which genomic regions will respond to *GLI3* perturbation.

Strikingly, we found that our Phospho-seq data for GLI3 could help predict which genomic regions would respond to perturbation. Examining the 5108 DARs identified in the perturbation experiment, we found that the accessibility of these sites was correlated to GLI3 protein levels in our Phospho-seq data, in contrast to >300,000 non-DAR ($p$-value = <0.001)(Fig. 5G). Together these results demonstrate that GLI3 does not act as a pioneer factor, but instead works with co-factors to regulate chromatin accessibility and gene expression as predicted by Phospho-seq. We conclude that Phospho-seq enables the

identification of putative co-factors and the discrimination of functionally regulated regions and genes.

## Using Phospho-seq data to identify gene regulatory networks in early neurodevelopment

By exploring relationships across modalities, multiomic datasets can help to reconstruct gene regulatory networks. While multiple groups have demonstrated how to leverage co-variation between chromatin accessibility and gene expression to link scATAC-seq peaks to the genes they regulate[14,49,57], our Phospho-seq data also allows us to link these peaks to transcription factors (TF) that regulate their accessibility. Recently, Argelaguet et al.[58] proposed in-silico ChIP-seq, a computational approach to predict TF binding events from multiomic data, based on co-variation between the gene expression of a transcriptional regulator and an scATAC-seq peak. We reasoned that we could extend this analysis using protein measurements from our Phospho-seq panel, which may be more reflective of TF levels, rather than RNA measurements.

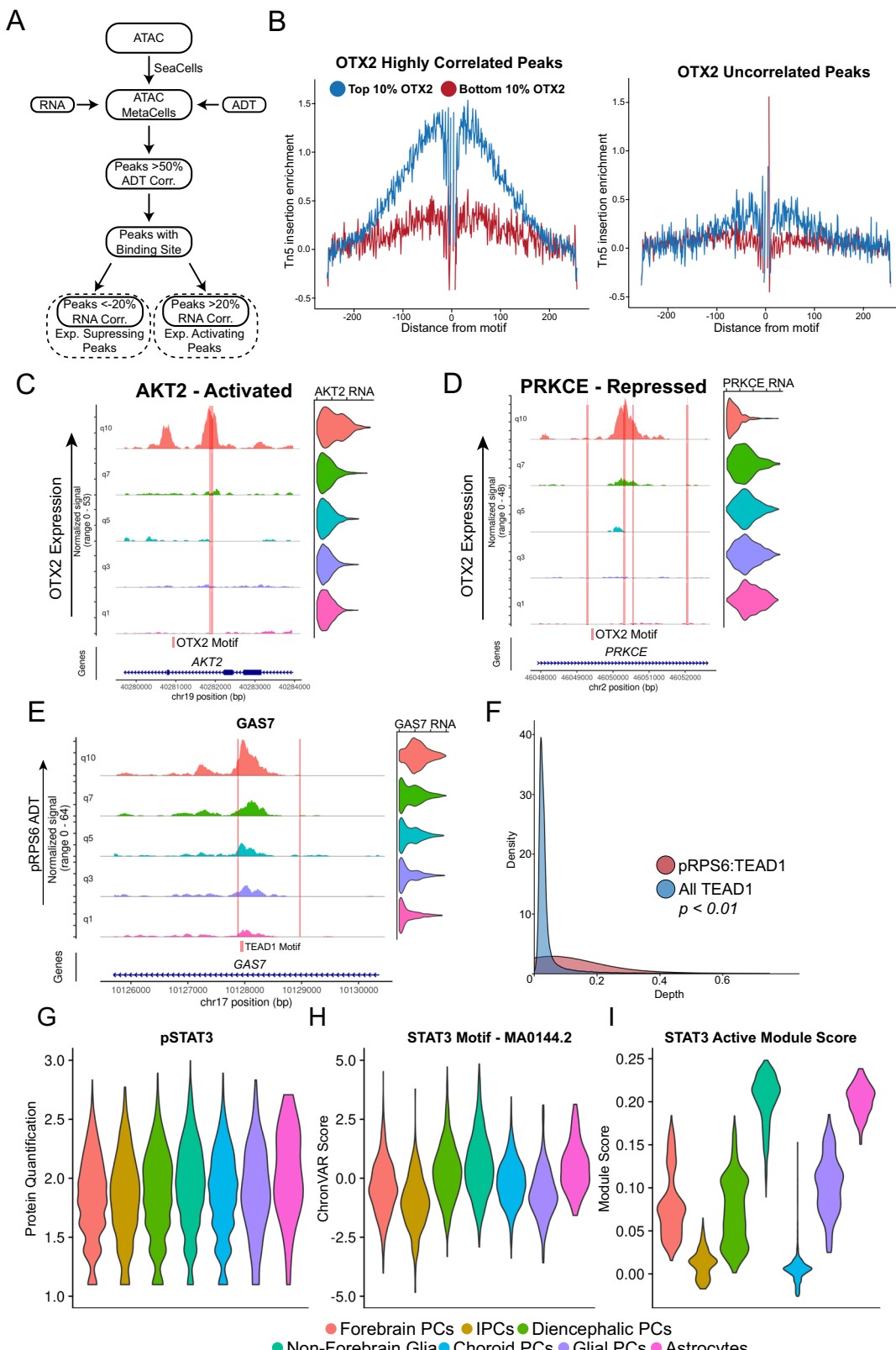

For each TF, we identified peaks harboring the binding motif and whose accessibility was highly correlated with TF protein abundance (Fig. 6A) using a high-resolution pseudo-bulking approach (SEAcells) to reduce the sparsity of scATAC-seq data[59] (Fig. S11A). Furthermore, we identified the subset of these peaks whose accessibility was correlated (or anti-correlated) with proximal gene expression. This procedure enabled identification of downstream activating and repressive

targets for individual TFs, as well as specific enhancer elements that are likely to mediate these relationships.

We applied this procedure across multiple neurodevelopmentally relevant TFs - OTX2, SOX2, NFIB, GLI3 and TBR2. Focusing on OTX2, we identified 5,033 linked peaks. When performing footprinting analysis, which aggregates accessibility profiles across the collection of linked peaks, we observed a depletion of accessibility centered at the exact TF

**Fig. 6 | Gene regulation inference with Phospho-seq. A** Schematic approach to use metacelling to discover cis-regulatory elements associated with individual proteins. **B** Tn5 cut-site footprinting between cells with high OTX2 expression and low OTX2 expression in peaks that are highly correlated with OTX2 (left panel) and uncorrelated with OTX2 (right panel). **C** Example of an inferred transcription activating peak associated with indicated proximal gene expression for OTX2. Coverage plots and violin plots are ordered by quantile ADT expression. Red lines indicated the location of an OTX2 binding motif. **D** Example of an inferred transcription repressing peak associated with indicated proximal gene expression for OTX2. Coverage plots and violin plots are ordered by quantile ADT expression. Red lines indicated the location of an OTX2 binding motif. **E** Coverage and violin plots of GAS7 split by quantiles of pRPS6 levels across the whole dataset with TEAD1 motifs indicated by red lines. **F** Density plot showing ChIP-Seq depth at all peaks with a TEAD1 motif compared to pRPS6:TEAD1 peaks identified in this study. The significance of $p = 1.9 \times 10^{-15}$ is determined by a Wilcoxon rank-sum test (two-sided). **G** Violin plot of normalized pSTAT3 ADT levels in a subset of progenitor and glial cells. **H** Violin Plot of STAT3 motif chromVAR scores in a subset of progenitor and glial cells. **I** Violin plot of STAT3 activated gene module scores in a subset of progenitor and glial cells.

binding site, which is indicative of TF binding (Fig. 6B and Fig. S11B, C). This footprint was specific only to cells expressing the TF, and to correlated peaks. For SOX2, further validation of predicted sites was performed using a ChIP-seq dataset[60] (Fig. S11D, E). We conclude that Phospho-seq enables the identification of bona fide transcription factor binding sites by leveraging correlations across multiple modalities at single-cell resolution.

Linking the candidate peaks to genes, we identified 3200 candidate targets (1772 activating, 1428 repressive) for OTX2 (Fig. 6C, D; Supplementary Data 4). Interestingly, we observed numerous cases where individual genes were regulated by multiple CREs, which were associated with different TFs. This included cases where putatively activating and repressive peaks were adjacently located, as is the case with OTX2 and SOX2 at the *OTX2-AS1* locus (Fig. S12A–C). When considering the additional TFs (Fig. S12D–G) and quantifying the overlap between activated and repressed gene lists, we identified some pairs of TFs, including SOX2 and GLI3, that regulate numerous overlapping genes in the same direction (activation or repression) and others, like OTX2 and SOX2, that act in opposing directions. (Fig. S12H).

We found that Gene Ontology (GO) enrichments for predicted target genes were consistent with each TF's known functional properties, including an enrichment for axon guidance amongst targets of TBR2 (a positive regulator of neuronal differentiation[61]), and an enrichment among SOX2 (a radial glial cell regulator[62]) targets for glial cell differentiation genes (Supplementary Data 6). For multiple TFs, we also observed an enrichment for genes associated with Wnt signaling, which is essential for brain patterning along the anterior-posterior axis. The pathway is known to increase activity in the posterior portion of the neural tube[63,64], supporting our observation of increased Wnt-related gene expression and increased chromVAR activities of Wnt-responsive TFs in diencephalic cells compared to forebrain (Fig. S13A). Furthermore, we observed a concordance in Wnt signaling with the activity of the Hippo pathway, which we associated earlier with MAPK/ERK signaling, providing further evidence, in concert with what has been shown[65], of the interconnectivity of signaling pathways in determining cell fate choice during early neurodevelopment (Fig. S13B). We conclude that our integrated Phospho-seq dataset can effectively identify relationships across modalities and reconstruct gene regulatory relationships.

### Linking phosphorylated proteins to gene regulatory elements with Phospho-seq

Using the pipeline above, we linked our previously observed pRPS6:TEAD1 relationship with specific cis-regulatory regions. We identified 481 putative TEAD-regulated CREs (Fig. 6E and Supplementary Data 4). GO analysis of activating gene links revealed enrichment for genes associated with cell division, growth, and morphogenesis, as would be expected for targets of both MAPK/ERK and Hippo (Fig. S14A and Supplementary Data 6). This suggests a model where activation of the Hippo pathway in neural cells leads to simultaneous phosphorylation of RPS6 through the interconnecting signaling pathways. If this model is true, there should be observable TEAD binding at the identified pRPS6:TEAD1 CREs. As our computational inference relies on Phospho-seq's predicted association between pRPS6-associated CREs

and binding of TEAD TFs, we had to validate this finding using ChIP-seq data quantifying TEAD1 binding[66,67]. We found that many of these pRPS6:TEAD1 peaks are observed to be bound by TEAD1 in cells of neural origin (Fig. S14B), with significantly more signal in this set of peaks compared to all genome-wide peaks with a TEAD1 motif (Fig. 6F). These results extend previously identified links between signaling pathways and TFs, originally identified in cancer cells, to cell state specification in neurodevelopment[68].

Finally, Phospho-seq can be used to measure phosphorylation among nuclear proteins, including transcription factors. Increased phosphorylation of the TF STAT3, which enhances its DNA-binding activity, is a characteristic feature of astrogliogenesis[69]. In our dataset, we confirmed a high-rank correlation of pSTAT3 with STAT3 motif accessibility, performing better than just STAT3 protein or RNA alone (Fig. S14C–E). We also saw an increase in pSTAT3 levels and STAT3 motif accessibility amongst glial cells in our dataset compared to progenitors or non-glial cells (Fig. 6G, H), confirming previous biological observations[70]. We then found 582 peak-gene links of CREs that harbored a STAT3 motif and whose proximal gene expression correlated with pSTAT3 levels. The expression of this gene set was specifically enriched in glial cells (Fig. 6I), and included canonical astrocyte markers such as *AQP4* (Supplementary Data 4). We conclude that by exploring relationships across multiple molecular modalities, Phospho-seq can highlight the role of cell signaling pathways in neurodevelopmental fate specification.

## Discussion

Here we present Phospho-seq, a scalable approach to detect intracellular and intranuclear proteins, including phospho-specific states, alongside additional molecular modalities. We confirmed the sensitivity and specificity of our approach in cell lines, and subsequently applied Phospho-seq to profile both retinal and brain organoids. We added transcriptomic information to these data using bridge integration, allowing us to maximize the data quality obtained from each modality. With this integrated dataset we linked protein expression levels to the accessibility of *cis*-regulatory elements and the expression of proximal genes. We demonstrate how these connections can assist in reconstructing gene regulatory relationships and provide insight to the causes and consequences of heterogeneous signaling during neurodevelopment.

We demonstrate that Phospho-seq is compatible with custom panels of user-conjugated antibodies, which can be easily and cost-effectively generated. While oligonucleotide-conjugated panels against cell surface proteins are readily available in both individual and pooled formats, these reagents are not widely available for intracellular targets, which has prevented the applicability of single-cell protein profiling to neuroscientific applications. Moreover, Phospho-seq enables the user to perform a multiplexed evaluation of the sensitivity and specificity for large panels of phospho-specific antibodies, by identifying correlations between ADT levels and chromatin and transcription states. We therefore expect that sequencing-based intracellular protein technologies will enable the identification and optimization of large intracellular panels, and that combining information across studies and biological systems will increase the generalizability of these results.

While pioneering technologies enable trimodal measurements of RNA, ATAC and surface protein or intranuclear abundance[16,18,20], in this study, we utilized bridge integration[23] to harmonize molecular modalities collected in separate scRNA-seq and Phospho-seq datasets. Although this approach requires running additional experiments, we and others have found that simultaneous profiling of ATAC and RNA in cells or nuclei reduces the data quality associated with each modality[49]. Improved fixation-compatible scATAC and scRNA methods may be helpful for addressing this issue in the future, but we note that a bridge-based experimental design may represent a flexible alternative for multimodal analysis where only a subset of samples need to be profiled with multiomic technologies. Importantly, we have previously shown that even small (<5000 cells) 10x Multiome datasets can serve as effective bridges for large unimodal datasets, as long as they are biologically representative[23]. Moreover, users of Phospho-seq may benefit from increasingly available reference atlases of either scRNA-seq or Multiome data[71,72], which would reduce the need for additional experimentation.

We anticipate that future studies will extend Phospho-seq to capture additional modalities related to chromatin state, and may shed additional light towards our understanding of how transcription factors regulate cellular chromatin. For example, combining Phospho-seq with scCUT&Tag[73] or NTT-seq[74] would identify TFs whose abundance correlated not only with chromatin accessibility, but with the presence of either activating or repressive chromatin marks. Further extensions that enable guide RNA capture (i.e. Perturb-ATAC[75], Spear-ATAC[76]) would enable multiplexed genetic screens to utilize Phospho-seq to perform massively parallel identification and characterization of signaling regulators. Finally, applying Phospho-seq in concert with spatially-resolved profiling technologies[77] may shed light on both intercellular and intracellular signaling networks. We hope that the broad applicability of Phospho-seq will facilitate its adoption in diverse biological contexts, including development, immunology, neuroscience, and cancer, to discover how cell signaling determines cellular behavior and fate.

## Methods

### Antibody selection

To select antibodies for use in the Phospho-seq assay, we used a hierarchical selection criteria based on available manufacturer's information. Antibodies were likely to be good candidates if they were first reported to work in intracellular flow cytometry and barring that, in immunocytochemistry on fixed cells. Antibodies that are reported to only work in western blot are less likely to work in Phospho-seq. Finally, the antibodies must be available in quantities of >5 µg in order to account for antibody loss in conjugation efficiency confirmation and post-conjugation processing. Antibodies from a variety of commercial vendors were used in this study including Abcam, Novus, R&D, ThermoFisher, Thomas Scientific, and BioLegend. A full list of antibodies and catalog numbers is available in Supplementary Data 1.

### Antibody purification

We used two methods of purifying antibodies from solutions with carrier proteins. For antibodies with BSA as a carrier protein, we used a BSA removal kit (Abcam: ab173231) following manufacturers instructions. For solutions that had more than just BSA contaminants (i.e. Ascites, gelatin), we used Magne Protein G beads (Promega: G7471). We followed manufacturer's instructions, eluting twice into a 100 mM Glycine-HCl solution (pH 2.7) and neutralizing with 2 M Tris-buffer (pH 7.5).

### Antibody conjugation and pooling

For antibody conjugation steps prior to purification, we followed the protocol described here with minor alterations (https://citeseq.files.wordpress.com/2019/03/cite-seq_hyper_conjugation_190321.pdf), which leverages a cost-efficient and flexible conjugation strategy originally developed for immuno-PCR[24]. Briefly, TotalSeq-style oligos were ordered from IDT with sequences of /5AmMC12/CCTTGGCACCCGAGAATTCCA XXXXXXXXXXXXXXXBAAAAAAAAAAAAAAAAAAAAAAAAAAAA*A*A for hashing, and /5AmMC12/GTGACTGGAGTTCAGACGTGTGCTCTTCCGAT CTNNNNNNNNNNXXXXXXXXXXXXXXXXXXNNNNNNNNNGCTTTAAGGCC GGTCCTAGC*A*A for antibody tags where X's represent the antibody barcode and N's represent UMI tags. We attached a TCO-PEG4 linker to each oligo through incubation with TCO-PEG4-NHS reagent (Click Chemistry Tools: A137-25) and then column purified them with Micro Bio-Spin 6 Columns (Bio-Rad: 732–6221), confirming labeling through observation of a size shift compared to unlabeled oligos on a 4% agarose gel. The concentration of each oligo was quantified and diluted to 100 µM. As ordered, ~50 nmoles of oligo can be used for labeling ~3 mg of antibodies, thus requiring the labeling process to only be performed once per oligo-tag sequence for hundreds of experiments.

Next, 5 µg or more of purified antibody was labeled with mTz-PEG4 (Click Chemistry Tools: 1069-10) for 30 min and conjugated to the TCO-PEG4 labeled oligo at a target concentration of 15 pmol per 1 µg of antibody overnight. This leaves each antibody conjugated to 2–4 copies of each oligo on average. Each conjugation was confirmed through running 1 µg of conjugated antibody on a 4–12% protein gel (Thermo: XP04125BOX) and observing a laddered size shift in increments of 30 kDa above the unconjugated IgG size of 155 kDa. The final concentration of each conjugated antibody was estimated through a BCA assay (Thermo: 23227).

Finally, for each experiment, 3–5 µg of each antibody was pooled together and treated with 40% saturated (~4.32 M) Ammonium Sulfate to salt out the conjugated antibodies from the unconjugated oligos still present in the antibody solution. After the majority of the remaining oligos were removed, the pooled solution was passed through a 50 kDa MWCO filter (EMD Millipore: UFC505024) 5–7 times —checking the flowthrough solution for ssDNA (leftover oligos) concentration after each centrifugation. When the flowthrough solution is at or below 0.2 ng/ul, the leftover antibody is resuspended from the filter and checked on a 4% agarose gel for two bands, one lower oligo band (90 nt) and one higher antibody band (~400 nt), with the aim of having the oligo band as faint as possible. The final antibody pool was then quantified through a BCA assay as there is an expected material loss (quantified up to 60% loss) of antibody bulk in this step due to the use of MWCO filters.

### Cell culture

K562 cells were maintained in IMDM media (ThermoFisher: 31980030) supplemented with 10% FBS (Corning:35-010-CV), 1 mM Pen-Strep (Sigma: P0781-50ML), and 1X Non-essential amino acids (Sigma:M7145-100ML). In experiments with stimulation and inhibition, 4 h before harvesting, K562 cells were removed from maintenance media, centrifuged for 5 min at $300 \times g$, and resuspended in IMDM media with no supplements. Cells were counted and $1 \times 10^6$ cells were transferred into two separate flasks. In one flask, cells were immediately treated with 500 nM PX-866 (Cayman Chemical: 13645) while in the other flask, 50 ng/ml EGF (R&D Systems: 236-EG-200) was added 15 min before harvest.

HEK cells were maintained in DMEM media (Caisson Labs: DML10) supplemented with 10% FBS, 1 mM Pen-Strep, and 1X Non-essential amino acids.

Three lines of iPSCs were obtained from the New York Stem Cell Foundation (51050,50828,50975). Cells were thawed into feeder-free conditions with mTESR1 (STEMCELL: 85850) and 10% CloneR (STEMCELL: 05888) with a Geltrex substrate (Fisher: A1413301). After two days, cells were changed into maintenance media – mTESR1 supplemented with 1 mM Pen−Strep. Throughout culture, media was changed daily and cell colonies were passaged by treatment with 1:3 Accutase (STEMCELL: 07920) diluted with PBS (Caisson Labs:PBL06-500ML) for 5 min. Cell lines were passaged a maximum of five times before an

experiment in order to reduce the number of cell-culture related chromosomal abnormalities.

## FACS

iPS cells were harvested through dissociation with accutase for 5 min and centrifuged in growth media at $300 \times g$ for 5 min. K562 cells in suspension were harvested through centrifugation in growth media. Both cell lines were resuspended in 1 ml PBS + 0.04% BSA (Fisher: BP9706-100) and run through a 40 μm filter to obtain a single cell suspension. These suspensions were further centrifuged for 5 min at $300 \times g$ in a tabletop swinging bucket centrifuge. Cells were resuspended in 450 μl of PBS and were fixed by adding 30 μl of 16% formaldehyde (Sigma: F8775-25ML) (final concentration, 1% FA in PBS). Cell suspensions were left to fix for 10 min at RT, with inversion every 3 min. The fixation reaction was quenched by adding 68.5 μl 1 M glycine and filling the tubes with ice-cold PBS. The suspensions were centrifuged for 5 min at $400 \times g$ at 4 °C, after which the supernatant was removed. The cells were resuspended in 1 ml PBS was added and the centrifugation was repeated. After the second centrifugation, the cells were resuspended in 100 μl of lysis buffer (10 mM Tris-HCl, 10 mM NaCl, 3.33 mM $MgCl_2$, 0.1% NP-40 (Thermo: 28324), 1% BSA in $H_2O$) and incubated on ice for 5 min for permeabilization. After 5 min, wash buffer #1 (10 mM Tris-HCl, 10 mM NaCl, 3.33 mM $MgCl_2$, 1% BSA in $H_2O$) was added up to 1 ml and cells were centrifuged for 5 min at $500 \times g$ after which the supernatant was discarded and the cells were resuspended in staining buffer (3% BSA in PBS) for blocking and placed in a tube rotator at RT for 30 min.

For cells that received a primary antibody bound by EcoRI single-stranded DNA binding protein (SSB) (Promega: M3011), the SSB binding reaction happened during the blocking step. Briefly, 1 μg of antibody was incubated with 8 μg of SSB in a solution of 1x NEB buffer 4 (NEB:B7004S) for 30 min at 37 °C. After this incubation, BSA and PBS were added to make a final solution of 3% BSA in 1X PBS.

For staining, the blocking buffer was removed through centrifugation at $600 \times g$ for 5 min and washing in wash buffer #2 (3% BSA in PBS + 0.1% Tween) three times. Cells were then resuspended in 100 μl of staining buffer with the appropriate antibodies and placed on the tube rotator for 1 h at RT. After this 1 h, the cells were washed three times in wash buffer #2 and then resuspended in staining buffer with a compatible secondary antibody (BioLegend: 406605, 407507, 406707, 408205) and stained for 30 min in the dark. Following three more washes, the cells were resuspended in MACS buffer (2 mM EDTA and 0.5% BSA in PBS) and run through the flow cytometry protocol on a Sony MH800 cell sorter with secondary only stained K562 cells acting as the negative control and a goal of 20,000 cells per condition. Flow cytometry analysis and plotting were performed using FlowJo v9.3.2.

For flow cytometry experiments with HEK and K562 cells, equal numbers of each of the cell types were processed in the same way as above.

For comparative experiments with ASAP-Seq and Phospho-seq, the cells were processed and stained as written below, but additionally stained during the primary staining step with a fluorescently conjugated primary pRPS6 antibody (BioLegend: 608603). Cells for each experiment were sorted into four bins representing quartiles (Fig. S2E). After sorting, the cells were hashed and processed (see below) and 3000 cells from each bin were sent into the 10x pipeline. The remaining cells for each bin were run through flow cytometry a second time using the pRPS6-PE fluorescence as a readout.

## Phospho-seq

The fixation and permeabilization steps for the Phospho-seq experiments proceeded the same as in FACS up until the blocking step, at which point in Phospho-seq, we also performed cell hashing[27]. For the pilot experiment, the iPS, K562-Stim, and K562-Inhib were separately stained with unique, home-conjugated hashing antibodies, while for the organoid experiments, the fixed permeabilized cells were split equally into four tubes for hashing. Briefly, 1 μg of TotalSeqA-conjugated CD298 and B2M antibody mix was added into the blocking buffer. Additionally, we included 100 μg of 30-nt 3′ blocked single-stranded DNA oligo to act as an additional block within the cells (NNNNNNNNNNNNNNNNNNNNNNNNNNNNNN/3ddC/). For the organoid experiments, we also included an additional 4 nM of four mixed unconjugated TSA oligos (free TotalSeq-A) that could be detected in each cell as a spiked-in control. The degree of free TotalSeq-A oligo helps to control for cellular differences in either the degree of permeabilization, or the degree of non-specific oligo binding. Cells were blocked for 30 min at room temperature on a tube rotator. During this blocking step, the primary antibody pool was incubated with SSB as above in preparation for primary antibody staining. For the pilot experiments, we used 0.5 μg/antibody for primary staining while for the retinal and brain organoid experiments we used 0.25 μg/antibody. After blocking, cells were centrifuged at $600 \times g$ for 5 min and washed once in wash buffer #2. Cells from each partition were resuspended in wash buffer #2, counted and combined in equal proportions up to 1 million cells total. Combined cells were centrifuged again at $600 \times g$ for 5 min, resuspended in the primary antibody solution and placed in a tube rotator for 1 h at RT. After primary antibody staining, cells were washed twice in wash buffer #2 and resuspended in 1x nuclei buffer (10×). The cell suspension was sent through a 40 μm filter and centrifuged at $600 \times g$ for 5 min. At this point, enough supernatant was removed to leave ~50 μl cell suspension. 5 μl of this cell suspension was used for quantification and the concentration was adjusted to 6000 cells/μl if possible. 5 μl of this solution (30,000 cells) was added to a solution of 7 μl ATAC buffer B and 3 μl ATAC enzyme (10x) and the rest of the protocol followed 10x scATAC kit v1.1 (10x Genomics: 100175) protocols with minor alterations in line with the ASAP-Seq protocol (below).

After tagmentation, 0.5 μl of 1 μM each of bridge oligo A and bridge oligo B was added to the barcoding reaction to allow for TotalSeq-A and TotalSeq-B oligo capture. To allow for better hybridization between the capture oligo, bridge oligo and antibody tags there was a 5 min 40 °C step added at the beginning of the GEM incubation. After silane bead elution (Step 3.1o), 43.5 μl instead of 40.5 μl Elution Solution I was added for elution−40 μl was kept for regular SPRI cleanup while 3 μl was set aside for added complexity in the tag library PCR. After the first SPRI cleanup (Step 3.2 d), the supernatant was saved instead of discarded. This supernatant contains all of the captured hashing and antibody tags, while the DNA on the beads contains the ATAC fragments and can proceed normally as written. The supernatant was treated with 32 μl of SPRI beads, allowed to bind for 5 min and subsequently cleaned up with 80% EtOH and eluted in 42 μl Buffer EB. This elute was combined with the 3 μl set aside from above and split into two tubes for amplification of TotalSeq-A and TotalSeq-B respectively. These two PCRs commenced separately with conditions as follows: 50 μl 2x KAPA HotStart ReadyMix (KAPA biosystems:07958935001), 2.5 μl 10 μM P5 primer (AATGATACGGCGAC CACCGA), 10 μM P7 primer (CAAGCAGAAGACGGCATACGAGATXXXXX XXXGTGACTGGAGTTCCTTGGCACCCGAGAATTCCA for TotalSeq-A and CAAGCAGAAGACGGCATACGAGATXXXXXXXXGTGACTGGAGTTCAGA CGTGTGC for TotalSeq-B, where Xs are i7 index nucleotides), 22.5 μl $H_2O$ and 22.5 μl input fragments. The thermal cycler program is: 95 C for 3 min, 14−18 cycles of 95 °C−20 s, 60 °C−30 s, 72 °C−20 s, then finally 72 °C for 5 min. These reactions were then cleaned up with 1.6X SPRI beads and eluted in 20 μl Buffer EB.

A step-by-step protocol can be found here: https://phospho-seq. com/post/protocol_2/PhosphoSeq_Protocol.pdf.

For the FACS-sorted benchmarking experiment, hashing was performed after sorting by incubating the cells within each bin in staining buffer with 1 μg hashing antibody, after which the cells were washed three times with 3% BSA in PBS + 0.1% Tween before inputting into the 10x scATAC-seq processing.

## ASAP-seq

For benchmarking experiments, ASAP-seq was performed as written[18] with intracellular staining buffer replaced by 3% BSA in PBS. Hashing was performed after sorting as detailed above. After pre-processing, all steps for ASAP-seq and Phospho-seq are identical.

## Sequencing and alignment

Assembled sequencing libraries were quantified by a Bioanalyzer High-Sensitivity DNA chip (Agilent: 5067-4626) and combined at a molar proportion of 25% ADT, 10% HTO, and 65% ATAC. These combined libraries were sequenced on a NextSeq 550 instrument using a 75-cycle high-output kit (Illumina: 20024906) with 34 bp Read 1, 8 bp i7, 16 bp i5 and 34 bp Read 2 parameters. ATAC Sequencing data were demultiplexed and aligned to the hg38 reference genome using CellRanger-ATAC v2.0.0 (10x genomics). ADT, HTO, and Free TotalSeq-A tag sequencing data were demultiplexed and aligned to index references using alevin[78]. Sequencing metrics for all experiments can be found in Supplementary Data 5.

## Data processing for pilot and benchmarking experiments

Chromatin data was processed with Signac v1.9[79]. We performed peak calling using the MACS3 algorithm[80], and performed downstream analysis using standard workflows (TFIDF, SVD, and LSI for dimensionality reduction) and default parameters. ChromVAR[28] was called within Signac for TF accessibility. HTOs were normalized using CLR normalization, and we performed demultiplexing and doublet detection using the HTOdemux function with default parameters. Only cells classified as singlets were kept, leaving 14,931 cells. Additional demultiplexing to assign cells to individual iPS donors was performed using Vireo v0.5.6[40], and relied on variant calling from 1000 Genomes Project and GnomAD common variants using Vartrix v1.1.22. ADTs were processed using default parameters from Seurat, including CLR normalization.

For benchmarking against FACS, staining indices for each protein tested were calculated according to the formula below[81]. Using the negative control sample, the median of the positive sample was subtracted from the median of the negative control samples and divided by two times the standard deviation of the negative control sample. This metric takes into account both the dynamic range of the assay, as well as the variation in the negative population.

$$\text{Staining Index} = \frac{\text{Positive Sample Median} - \text{Negative Sample Median}}{2 \times (\text{SD Negative Sample})}$$

(1)

## Retinal organoid culture and harvesting

To produce retinal organoids, we used the 01F49i-N-B7 (B7) iPS cell line from the Institute of Molecular and Clinical Ophthalmology Basel, which has been previously shown to reliably produce retinal organoids[31]. The organoids were generated using the AMASS protocol[31] exactly as used in Wahle, Brancati, Harmel, and He[32]. iPSCs were cultured in mTeSR1 (STEMCELL Technologies, 85857) or mTeSR Plus (STEMCELL Technologies, 100−0276) supplemented with penicillin−streptomycin (1:200, Gibco, 15140122) on Matrigel-coated plates (Corning, 354277). Cells were split 1–2 times per week after dissociation with EDTA in DPBS (0.5 mM) (Gibco, 12605010). Media were supplemented with rho-associated protein kinase (ROCK) inhibitor Y-27632 (5 μM, STEM- CELL Technologies, 72302) after thawing. To generate retinal organoids, 600 cells were seeded in each microwell of an agarose chamber (MicroTissues 3D Petri Dish micro-mold, Z764019). At the time points of interest, organoids of the same batch were pooled and dissociated. Organoids were washed three times with HBSS without Ca²⁺/Mg2+ (STEM- CELL Technologies, 37250). Single-cell suspensions were obtained using a papain-based dissociation kit

(Miltenyi Biotec, 130-092-628). In brief, 1 ml of pre-warmed papain solution was added to the organoids and incubated for 10 min at 37 °C. To facilitate dissociation, the mix was pipette-mixed every 5 min with a p1000. Enzyme mix A was added and mixed by inversion and incubated for 10 min at 37 °C. Samples were pipette-mixed until tissue dissociation was confirmed via visual inspection. After incubation, cells were centrifuged for 5 min at 300 g and 4 °C. The cells were then resuspended in 250 μl of PBS + 0.04% BSA and sequentially filtered through a 70-μm filter (pluriSelect Mini, 43-10070-50) and a 40-μm filter (pluriSelect Mini, 43-10040-40) and subsequently cryopreserved in CryoStor CS10 (STEMCELL Technologies: 07952) at −80 °C before moving to liquid nitrogen storage the following day.

On the day of the experiment, cells were removed from cryopreservation and thawed at 37 °C. Cells were deposited into Neurobasal media (Thermo: 21103049) supplemented with 5% FBS and centrifuged for 5 min at 300 × g. Cell pellets were resuspended in 450 μl PBS, and run through a 40 μm filter, after which fixative was added and the Phospho-seq experiment continued as written above.

## Initial data processing for retinal organoid experiment

Sequencing and data processing were performed as detailed in the pilot experiment. We removed both cells with low sequencing depth for either the ADT or ATAC as well as doublets defined by hashing and HTOdemux. This left 8136 cells in total in the final dataset. The ADT signal was normalized using scTransform v2[82]. When running scTransform, we regressed against the number of free TotalSeq-A counts, which represent the degree of cell permeabilization. This allows us to control for cellular variation in the degree of both, and/or higher degrees of intracellular stickiness for free oligo, both of which are reflected in the single-cell quantifications of free oligo[83,84]. While we recommend this procedure in general for Phospho-seq experiments, standard CLR normalization yields very similar values, and does not modify our biological conclusions.

## Immunofluorescence for retinal organoid experiment

To validate Phospho-seq observations, a small iterative indirect immunofluorescent imaging (4i) experiment with two imaging rounds was performed on week 39 retinal organoids using identical techniques and samples as Wahle, Brancati, Harmel, and He[32]. The antibodies used in this experiment are listed in Supplementary Supplementary Data 1.

## Phospho-seq-multi on retinal organoids

To run Phospho-seq-multi, retinal organoids were cultured as detailed above. Organoids aged 18 and 24 weeks were harvested separately and processed in parallel for the initial Phospho-seq-multi steps including fixation, permeabilization, blocking, and hashing before being combined at the primary staining step. The processing for the Phospho-seq-multi protocol is identical to the Phospho-seq protocol with the following changes, as inspired by DOGMA-seq[18]. First, NxGen RNAse Inhibitor (Lucigen: 30281-1) was added at 1 U/μl for all wash buffers and at 2U/μl for permeabilization and staining buffers. Correspondingly, DTT (Invitrogen: y00147) was added to these buffers at 1 mM final concentration. Next, the 10x Multiome kit (10x Genomics: 1000285) was used in place of the scATAC kit. To facilitate the capture of TSB-tagged ADTs, Bridge Oligo B (0.5 μl of 1 μM) was spiked into the barcoding master mix (Step 2.1a). Bridge Oligo A spike-in was not necessary due to the direct poly-A capture of TSA oligos in the multiome procedure. As there is a 37 °C step immediately following GEM generation, the additional 5 min 40 °C incubation as in Phospho-seq-ATAC is also not necessary here. From there, the protocol proceeds as written until the preamplification PCR (Step 4.1) where 1 μl of 0.2 μM HTO additive (CCTTGGCACCCGAGA ATT*C*C) and ADT additive (GTGACTGGAGTTCAGACGTGTGC*T*C) oligos were added to the master mix to amplify the TSA and TSB oligos. After preamplification and SPRI cleanup (Step 4.3k), the products are

eluted in 100 µl buffer EB as opposed to 160 µl as written. The ATAC library is amplified using 25 µl (25%) elute as the input with 15 µl H$_2$O. The cDNA library is amplified using 35 µl (35%) of the elute as input and then the ADTs and HTOs are amplified using 20 µl (20%) each of the elute inputs combined with 25 µl H$_2$O. The ATAC and cDNA amplification proceed as written in the protocol and the ADT and HTO amplifications proceed as written above for Phospho-seq using the same P7 primers but with the SI-PCR P5 primer (AATGATACGGCGACCACCGAGATCTACACTC TTTCCCTACACGACGC*T*C). A step-by-step protocol can be found here: https://phospho-seq.com/post/protocol_3/PhosphoseqMulti_Protocol.pdf

### Phospho-seq-multi sequencing and alignment
Libraries were quantified using a Bioanalyzer as above and sequenced on a NextSeq 550 instrument. The cDNA library was sequenced using a 75-cycle high-output kit (Illumina: 20024906) with 40 bp Read1, 8 bp Index i7, and 44 bp Read 2. The ATAC and ADT libraries were sequenced using a custom sequencing recipe with a 150-cycle mid-output kit (Illumina: 20024904) with 50 bp Read 1, 8 bp Index i7, 86 bp Read 2, and 16 bp Index i5 (with 8 dark cycles). ATAC and cDNA libraries were aligned and processed using CellRanger-arc 2.0.0 to the hg38 genome. ADTs were processed using Alevin as above.

### Initial data processing for Phospho-seq-multi experiment
Cells were processed as above, while removing outliers with either >10,000 or <100 RNA UMIs and >100,000 or <1000 peaks. This left 1474 cells in this dataset with all other processing remaining the same.

### Bridge integration on retinal organoids
Previously processed and published scRNA reference datasets for Week 38 and scMultiome bridge for Week 36 retinal organoids were obtained from public sources[32]. The reference dataset had 22,011 cells and the bridge dataset had 4926 cells. Bridge integration and RNA imputation for the Phospho-seq dataset were performed using the standard bridge integration workflow[23] described here: https://satijalab.org/seurat/articles/seurat5_integration_bridge. First, we re-called peaks in the Phospho-seq ATAC dataset using only those peaks present in the bridge dataset. This procedure identifies 'anchors' between the Phospho-seq and scRNA-seq datasets, after representing them both based on the same multi-omic dictionary. After the transfer anchors were determined, the cell labels and RNA modality were transferred onto the Phospho-seq dataset using the TransferData function in Seurat.

### Brain organoid culture and harvesting
We used four human iPS cell lines (Hoik1 and Wibj2 from the HipSci resource[85], 01F49i-N-B7 (B7) from the Institute of Molecular and Clinical Ophthalmology Basel, and WTC from the Allen Institute). Stem cell lines were cultured in mTESR1 and supplemented with penicillin–streptomycin (1:200, Gibco: 15140122) on Matrigel-coated plates (Corning: 354277). Cells were passaged 1–2 times per week after dissociation with TryplE (Gibco: 12605010) or EDTA in DPBS (final concentration 0.5 mM) (Gibco:12605010). The medium was supplemented with Rho-associated protein kinase (ROCK) inhibitor Y-27632 (final concentration 5 µM, STEMCELL Technologies: 72302) the first day after passage. Cells were tested for mycoplasma infection regularly using PCR validation (Venor GeM Classic, Minerva Biolabs) and found to be negative. A total of 2000–3000 cells were aggregated in ultra low-attachment plates (Corning: CLS7007) to generate brain organoids using a whole-brain organoid differentiation protocol[86].

Single-cell suspensions were acquired by dissociation of the organoids with a papain-based dissociation (Miltenyi Biotec: 130-092-628). The organoids were cut into pieces and washed with HBSS (no magnesium, no calcium, Gibco: 14170120) to remove debris before prewarmed papain solution (2 ml) was added and incubated for 15 min

at 37 °C. Enzyme mix A was added before the tissue pieces were triturated 5–10 times with 1000 µl wide-bore and P1000 pipette tips. The tissue pieces were incubated twice for 10 min at 37 °C with trituration steps in between and after with P200 and P1000 pipette tips. Cells were filtered consecutively with a 30 µm pre-separation filter and centrifuged. Cells were counted and 1–3 million cells pre-sample were cryopreserved in CryoStor CS10 (STEMCELL Technologies: 07952) at −80 °C before moving to liquid nitrogen storage the following day.

On the day of the experiment, cells were processed and sequenced identically to the retinal organoids.

### Initial data processing for brain organoid experiment
Chromatin data was quantified, normalized, and processed using the same workflow as above. We removed one cluster of cells that was defined by low sequencing depth. We assigned each cell to its organoid line and donor-of-origin using Vireo v0.5.6[40], and relied on variant calling from 1000 Genomes Project and GnomAD common variants using Vartrix v1.1.22.

For ADT processing, cells with over 10,000 ADT counts were removed from the analysis, as these were outliers in ADT data. This left 9,034 cells total in the final dataset. ADT signal was normalized using scTransform v2[82] and regressed against the number of free TotalSeq-A counts as above. To identify relationships between ADT levels and TF activity scores, we calculated Pearson correlations between each ADT value and each TF motif chromVAR scores across all cells. These correlations were then plotted according to their rank compared to every other TF motif in the JASPAR2020 motif dataset.

### Brain organoid bridge integration
Additional single-cell experiments were performed for bridge integration on the same Day 90 organoid samples. For the bridge dataset, nuclei were extracted from cells and run through the 10X Multiome ATAC + Gene Expression protocol according to manufacturer's instructions (10X Genomics:1000285). Libraries were sequenced on a NextSeq 550 instrument using a 150-cycle high-output kit with 50 bp R1, 8 bp i7, 16 bp i5, and 86 bp R2. Data was aligned using CellRanger-arc v 2.0.0 to hg38. Aligned data was processed using default parameters in Seurat and Signac including a cutoff of 500 counts and SCT normalization for the RNA modality. Nuclei were demultiplexed using variant calling on the ATAC modality and only singlet nuclei were used. This left 4958 nuclei in the bridge dataset.

For the whole cell scRNA-Seq dataset, whole, unfixed cells were hashed with four homemade hashing antibodies, loaded at 25,000 cells/lane, and run through the 10x scRNA-Seq v3.1 kit according to manufacturer's instructions (10x Genomics: 1000268). Separate HTO and RNA libraries were constructed and sequenced on a NextSeq 550 instrument at a ratio of 90% RNA and 10% HTOs using a 75-cycle high-throughput kit with 28 bp R1, 8 bp i7, and 56 bp R2. Transcriptomic data were aligned using CellRanger v7.0.0 to hg38 and HTO data was aligned to indices using Alevin. Cells were processed using default parameters in Seurat with SCT normalization. HTOs were quantified and cells were clustered according to HTO expression, with clusters representing doublets removed from subsequent analysis. This processing resulted in 19,280 cells in the final reference dataset.

Bridge integration and RNA imputation for the Phospho-seq dataset were performed using the standard bridge integration workflow as above.

### Pseudotime and trajectory analysis
To look for gene expression, protein, and chromVAR patterns across a trajectory, cells were first subset by both donor and cell type lineage - Hoik1 cells and forebrain neuronal lineage or B7 cells and diencephalic lineage. The destiny[42] package in R was used to create diffusion maps based on the lsi embeddings from these subset datasets. Monocle3[41] was then run on the diffusion maps to calculate

pseudotime, setting the cell with the minimum value in diffusion component 1 as the origin.

## Immunofluorescence on brain organoids

Organoids were fixed overnight in 4% PFA. The next day, the organoids were washed three times for 5 min with DPBS and then transferred into 30% sucrose in DPBS until they sank to the bottom of the tube. Following the organoids were transferred into cryomolds (Sakura, #4565) and embedded in Tissue-Tek O.C.T. (Sakura, #16-004004) on dry ice. The organoids were then sliced at the Cryostar NX70 (Thermo Scientific) into 20 μm thick slices at −17 °C. The slices were transferred to SuperFrost® Plus glass slides and washed with PBS. After a quick wash, antigen retrieval was performed for 20 min at 70 C in 1x preheated HistoVT one (Nacalai, #06380). Slides were washed three times for 5 min with PBS + 0.2% Tween and then transferred to blocking and permeabilization (PBS, 0.1% Triton, 5% Serum, 0.2% Tween, 0.5% BSA) for 1 h. The antibodies were added in blocking solution overnight at 4 °C. The next day the slides were washed again three times for 5 min with PBS + 0.2% Tween and the secondary antibody was added to the blocking solution for 2 h at room temperature. Last, the slides were washed again three times for 5 min with PBS + 0.2% Tween adding DAPI to the last wash. The slides were then mounted with Prolong Glass Antifade and imaged using a confocal Zeiss LSM980 microscope (Airyscan).

## Identifying relationships between ADT and cis-regulatory elements

Before linking scATAC-seq peaks to putative protein levels, we first ran the SEACells algorithm to increase the robustness of the chromatin data. We ran SEACells using the suggested parameters of 75 cells per metacell (120 metacells for the Phospho-seq dataset), n_waypoint_eigs = 10 and waypoint_proportion = 0.9. Based on the SEACells grouping, we calculated cumulative count values from the ATAC, ADT, and RNA modalities. All three modalities were normalized after aggregation: ADT using CLR normalization, ATAC using TF-IDF and RNA using log-normalization.

To determine candidate peaks associated with ADTs that show regulatory potential for proximal genes, we first found peaks that were correlated to each protein. For this we ran the Pearson correlation between each peak and ADT within the metacell object. We first filtered the peaks with a correlation of >0.5, which for SOX2 and OTX2 represents the top ~2.5% of correlated peaks. For ADTs without any peaks with >0.5 correlation, we reduced the correlation cut-off by 0.1 until we observed >1% of peaks correlating. This resulted in a correlation cut-off of 0.4 for TBR2 and 0.3 for pRPS6 and pSTAT3. We next filtered these peaks for those that harbored the corresponding TF binding motif as indicated by the JASPAR database, using a probability score cutoff of >400. Peaks that passed both the correlation-based and sequence-based criteria were linked to the associated ADT.

We next aimed to link these peaks to putative target genes. To do this, we used the LinkPeaks function in Signac with parameters pvalue_cutoff = 0.10, score_cutoff = 0.01. We only considered gene-peak links that fell within 500 kb. For each identified gene-peak link, we calculated the Pearson correlation across meta-cells between peak accessibility and gene expression, and retained all peaks with correlation >0.2 (activating) or <−0.2 (repressive). These cut-offs represented approximately the top 10% positively and negatively correlated genes with ADT expression for each ADT. The lists of each of these peak-gene relationships are in Supplementary Data 4.

## Processing additional GLI3 data

To process previously published[37] GLI3 Cut & Tag data performed on Week 3 brain organoids, we first aligned raw FASTQ files to the human genome using Bowtie2. The alignment reads were then deduplicated using Picard v2.8 and peaks representing GLI3 bound fragments were called using Genrich v0.6 using default settings.

Next, we obtained previously published multiome data from Week 3 GLI3 KO and GLI3 WT organoids. Since the CUT&Tag dataset was performed on a bulk sample, we used all cells in the multiome dataset. Aiming to focus on direct regulation, we quantified a cell-peak accessibility matrix using the previously identified GLI3 CUT&Tag binding sites. We identified differentially accessible sites between wildtype and knockout organoids using the FindMarkers function in Signac. Motifs enriched in both differentially accessible and non-differentially accessible regions were discovered using the FindMotifs function in Signac, which identifies enriched motifs in a peak set against a set of matching background peaks. Links between GLI3 protein expression, ATAC-seq peaks, and downstream peaks were calculated as described above.

## TF footprinting

For TF footprinting analysis, motifs were first identified using the Signac function AddMotifs with JASPAR2020 motif dataset. We only considered motifs within called peaks and then split the peaks into categories based on correlation between peak accessibility and TF protein expression. We selected highly correlated peaks in the top third of expression and a second set of peaks with negative correlation. The Footprint function in Signac was used to sum Tn5 insertion events surrounding motif instances in the highly correlated and negatively correlated peaks. We partitioned cells into ten quantiles based on TF protein expression and displayed Tn5 insertion enrichment for the first and tenth expression quantile using the PlotFootprint function in Signac.

## ChIP-Seq benchmarking

For benchmarking our SOX2 candidate peaks, we relied upon a previously published SOX2 ChIP-Seq dataset performed on in vitro differentiated human neural progenitor cells (hNPCs) and human embryonic stem cells (hESCs)[60]. We plotted the ChIP-Seq data alongside our ATAC-seq data by converting the raw data into a BigWig format and plotting with the CoveragePlot function in Signac. Additional benchmarking was performed by quantifying the depth of SOX2 signal across 1180 SOX2 candidate peaks - those with correlated chromatin accessibility with SOX2 ADT, possessing a SOX2 binding motif and high correlation for proximal RNA, as outlined in Fig. 6. We repeated this analysis for all 31,782 peaks that possessed a SOX2 motif, excluding those correlated peaks and compared the two signals using a Wilcoxon rank-sum test. For benchmarking TEAD1:pRPS6 candidate peaks, data from a TEAD1 ChIP-seq experiment in glioblastoma cells[66] was downloaded and quantified in the same way as the SOX2 ChIP-seq dataset across all peaks in our dataset with a TEAD1 motif as well as the 481 TEAD1:pRPS6 peaks that we identified. For visualization, the data along with another TEAD1 ChIP dataset from K562 cells[66] was converted to a BigWig file and plotted as above.

## Gene ontology analysis

The genes associated with the top 500 peak-gene links for each TF were used to analyze gene ontology category enrichment using the enrichr R package[87]. For instances where there were fewer than 500 peak-gene links, all the peaks were used. The GO categories were then filtered to include only those with ≥5 genes as hits.

## Glial subsetting and reference mapping

Cells that mapped to glial cell identities in the Phospho-seq dataset were subset and reclustered. These clusters were assigned their described identity based on previously reported glial markers[88,89]. To determine regional identity of these clusters, cells of class Glioblast, Radial Glia and Oligo from an atlas of first trimester developing brain[52] were reference mapped against these clusters and cells with high mapping scores for any cluster were subset and analyzed for gene expression of marker genes and regional identity from the metadata.

## STAT3 active module score

To calculate and plot the STAT3 active module score per cell, we extracted the gene names for all peak-gene links associated with high pSTAT3 signal and increased proximal gene expression. We then used the AddModuleScore() function in Seurat to calculate the collective expression of those genes from the RNA modality in each cell.

## Data availability

The processed Phospho-seq data are available at: https://zenodo.org/record/7754315. The sequencing data generated in this study have been deposited in the GEO database under accession code GSE285561.

## Code availability

Seurat and Signac are freely available as open-source software packages at: https://github.com/satijalab/seurat. https://github.com/stuartlab/signac.

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

## Acknowledgements

The authors would like to thank all the members of the Satija Lab and Treutlein Labs for thoughtful discussions related to this work, and

specifically Sophie Jansen in the Treutlein Lab. J.D.B. is a postdoctoral fellow of the Jane Coffin Childs Memorial Fund for Medical Research. This work was supported by the Chan Zuckerberg Initiative (EOSS-0000000082, HCA-A-1704-01895 to R.S.), and the NIH (RM1HG011014-02, 1OT2OD026673- 01, DP2HG009623-01, R01HD096770, R35NS097404 to R.S.). B.T. was supported by the European Research Council (758877-Organomics), the Swiss National Science Foundation (Project Grant-310030_192604), and F.Z. was supported by EMBO Long-Term Fellowship ALTF 36-2021.

## Author contributions

J.D.B. and R.S. conceived of the study and wrote the manuscript. J.D.B. developed the Phospho-seq technology with intellectual guidance by R.S. J.D.B., and C.D. generated the flow cytometry and sequencing data. P.W., G.B., and F.Z. grew the organoids and generated the immuno-fluorescence data under the supervision of B.T. J.D.B. and A.H. performed the data analysis.

## Competing interests

In the past three years, R.S. has worked as a consultant for Bristol-Myers Squibb, Regeneron, and Kallyope and served as an SAB member for ImmunAI, Resolve Biosciences, Nanostring, and the NYC Pandemic Response Lab. The other authors declare no competing interests.
