## [Transparent Peer Review file · Nature Communications]

Phospho-seq: Integrated, multi-modal profiling of intracellular protein dynamics in single cells

Corresponding Author: Dr Rahul Satija

Version 0:

Reviewer comments:

Reviewer #1

(Remarks to the Author)

All of my comments have been addressed.

Reviewer #3

(Remarks to the Author)

In their revised manuscript the authors have addressed most of the main criticisms of this reviewer satisfactorily, especially the inclusion of details on the antibody detection using the phospho-seq procedure. For instance, the inclusion of the staining index will allow the readers to gauge the performance and signal to noise ratio (assuming the sample with the lowest value is indeed 'noise') of individual antibodies. This has significantly improved the manuscript for that aspect and I commend the authors for adding this information to enable readers to gauge the quality of the data generated using phospho-seq.

That said, there are some remaining issues and/or comments on the newly added analyses/figures that are discussed below.

1. The authors now included details on the antibody detection of all the experiments in Table S5. While the metrics for the phospho-seq experiments look convincing. For instance, the number of ADT UMI counts per cell is solid, as is their ratio with the mean number of sequencing reads per cell (as an indication of ADT library complexity). In contrast, the mean ADT UMI count per cell for the phospho-seq multi is extremely low (39 counts from a mean sequencing depth of >12,000 reads per cell). This really indicates that the phospho-seq multi does not yield usable ADT data. Therefore, the conclusion of the authors that phospho-seq is amenable to experimental trimodal profiling is really not supported (line 328).

Although it is fully understandable that the authors would want to include work that probably took a lot of effort and resources in a manuscript, it is not clear (considering the low quality of the ADT component) what the added benefit of including the phospho-seq multi experiment and analyses is for the manuscript. Considering that the authors, rightly, continue with using bridge-integrated data of much better quality, the flow of the manuscript does not require including phospho-seq multi per se either.

2. New Figure S2G-H on the comparison between phospho-seq and ASAP-seq: It is unclear what the Fold Change represents and how this is represented. Is this log₂ FC? If so, it seems somewhat counterintuitive from the pRPS6-SE signal levels in Figure S2E that this Fold Change between the first and 4th bin should be so modest (maximum of the range is 0.2). Moreover, it is good to mention that (as with most/many sequencing-based measurements) there is ~50% compression of the fold changes as determined with phospho-seq compared to Flow.

3. The authors state that 'most ADTs had high correlation with their corresponding imputed RNA values (50% > 80th percentile rank-correlation) (Fig S6C).

Without the intention of being pedantic: a high rank correlation is not the same as a high correlation. The actual correlation values, as estimated from the presented figures, are generally $R < 0.5$. Can this be claimed to be high?

This is an example of the type of phrasing that was pointed out in the original review (although this particular claim was not explicitly highlighted) that represents stronger claims than perhaps warranted by the data. I urge the authors to be careful and not overreach conclusions.

4. On line 512 the authors state: “we identified synergistic relationships between some pairs of TFs, including SOX2 and GLI3, as well as antagonistic relationships between others (OTX2 and SOX2)(Figure S12H). This figure represents a heatmap of log transformed Jaccard similarities (details are missing but it is presumably log base 2?). Although the heatmap does show clear grouping in this representation, the underlying Jaccard similarities may be quite low (<20-25% overlap between peaks of TFs it seems). It is therefore very difficult to know what the claim of synergy (or antagonism) is based on. Usually claims about synergy are made on quantitative analyses and showing some degree of overlap to would not distinguish between synergy and additive effects for instance.

5. Some of the references to the supplemental figures are incorrect and should be corrected.

signed:
Klaas Mulder

Point by Point Reviewer Response:

We would like to thank the reviewers for their insightful comments and for helping to improve the manuscript. In response to their concerns, we have added numerous clarifications and further expanded supplementary figure 2.

REVIEWERS' COMMENTS:

Reviewer #1 (Remarks to the Author):

All of my comments have been addressed.

We appreciate this reviewer's feedback and would like to thank them for their prompt assessment.

Reviewer #3 (Remarks to the Author):

In their revised manuscript the authors have addressed most of the main criticisms of this reviewer satisfactorily, especially the inclusion of details on the antibody detection using the phospho-seq procedure. For instance, the inclusion of the staining index will allow the readers to gauge the performance and signal to noise ratio (assuming the sample with the lowest value is indeed 'noise') of individual antibodies. This has significantly improved the manuscript for that aspect and I commend the authors for adding this information to enable readers to gauge the quality of the data generated using phospho-seq.

Thank you very much for the kind words and we appreciate the reviewer's guidance in improving the manuscript.

That said, there are some remaining issues and/or comments on the newly added analyses/figures that are discussed below.

1. The authors now included details on the antibody detection of all the experiments in Table S5. While the metrics for the phospho-seq experiments look convincing. For instance, the number of ADT UMI counts per cell is solid, as is their ratio with the mean number of sequencing reads per cell (as an indication of ADT library complexity). In contrast, the mean ADT UMI count per cell for the phospho-seq multi is extremely low (39 counts from a mean sequencing depth of >12,000 reads per cell). This really indicates that the phospho-seq multi does not yield usable ADT data. Therefore, the conclusion of the authors that phospho-seq is amenable to experimental trimodal profiling is really not supported (line 328).

Although it is fully understandable that the authors would want to include work that probably took a lot of effort and resources in a manuscript, it is not clear (considering the low quality of the ADT component) what the added benefit of including the phospho-seq multi experiment and analyses is for the manuscript. Considering that the authors,

rightly, continue with using bridge-integrated data of much better quality, the flow of the manuscript does not require including phospho-seq multi per se either.

We appreciate the reviewer's feedback regarding this point. We agree that the ADT detection in the Phospho-seq Multi is highly inefficient, however, despite this, there was a concordance between chromatin accessibility and protein levels (See Supplementary Figure 4B) and canonical marker ADTs were still detected in the expected cell types (See Figure 3C) indicating that the technique does work, just not well, as is stated in the text. Furthermore, this experiment was initially performed in response to initial reviewer feedback, so we fear by taking it out, it would weaken the manuscript and also, by omission, encourage readers to attempt to apply the multiome to the Phospho-seq workflow without being aware of its drawbacks. Additionally, we postulate that a source of inefficiency is the use of Total-seq-B ADTs instead of Total-Seq-A ADTs, as in NEAT-Seq (Line 278) (Chen, et al, *Nat Methods*, 2022), helping to guide future users if they would like to further troubleshoot the technique. Given these points, we will keep this experiment and data within the manuscript but we have changed the language on the indicated lines above to further reflect its inefficiencies.

2. New Figure S2G-H on the comparison between phospho-seq and ASAP-seq: It is unclear what the Fold Change represents and how this is represented. Is this log₂ FC? If so, it seems somewhat counterintuitive from the pRPS6-SE signal levels in Figure S2E that this Fold Change between the first and 4th bin should be so modest (maximum of the range is 0.2). Moreover, it is good to mention that (as with most/many sequencing-based measurements) there is ~50% compression of the fold changes as determined with phospho-seq compared to Flow.

Thank you for highlighting this point. For this experiment, the cells were sorted into four bins and then each of those bins were used for a sequencing experiment (Phospho-seq or ASAP-Seq) as well as an additional flow cytometry experiment. This is an experimental framework adapted from the initial CITE-seq paper (Stoeckius et al, *Nature Methods*. 2017; Fig 2E-F). The intent for performing it this way was to control for the noise from sorting so that flow cytometry and sequencing would be compared from the same post-sorting population of cells. Thus, the fold-change calculated for the flow cytometry measurements is between the bins after they were initially sorted and Supplementary Figure 2 has been updated to add more information regarding this aspect of the experiment. In response to the first point, the scatter plots have been updated to reflect log₂ fold-changes on both axes for consistency across the manuscript.

3. The authors stat that 'most ADTs had high correlation with their corresponding imputed RNA values (50% > 80th percentile rank-correlation) (Fig S6C). Without the intention of being pedantic: a high rank correlation is not the same as a high correlation. The actual correlation values, as estimated from the presented figures, are generally $R < 0.5$. Can this be claimed to be high? This is an example of the type of phrasing that was pointed out in the original review (although this particular claim was not explicitly highlighted) that represents stronger

claims than perhaps warranted by the data. I urge the authors to be careful and not overreach conclusions.

Thank you for bringing this point up, we agree with the reviewer and apologize for this oversight. The language has been amended to reflect and emphasize that this is a rank-correlation measurement and not a true correlation measurement.(Lines 237-241;-345-347)

4. On line 512 the authors state: “we identified synergistic relationships between some pairs of TFs, including SOX2 and GLI3, as well as antagonistic relationships between others (OTX2 and SOX2)(Figure S12H).

This figure represents a heatmap of log transformed Jaccard similarities (details are missing but it is presumably log base 2?). Although the heatmap does show clear grouping in this representation, the underlying Jaccard similarities may be quite low (<20-25% overlap between peaks of TFs it seems). It is therefore very difficult to know what the claim of synergy (or antagonism) is based on. Usually claims about synergy are made on quantitative analyses and showing some degree of overlap to would not distinguish between synergy and additive effects for instance.

Thank you for this comment – we understand and agree about the point regarding specificity in language identifying synergistic peaks versus additive CREs versus antagonistic CREs and have updated the language to be less definitive regarding mechanism of action, while still highlighting the overlapping direction of action between pairs of TFs (lines 481-484).

5. Some of the references to the supplemental figures are incorrect and should be corrected.

Thank you for pointing this out, the figure order has been amended to ensure that all figure call outs are to the intended figure panel.